

# Multi-observational estimation of regional and sectoral emission contributions to the persistent high growth rate of atmospheric CH₄ for 2020–2022

Yosuke Niwa[1], Yasunori Tohjima[1], Yukio Terao[1], Tazu Saeki[1], Akihiko Ito[2], Taku Umezawa[1], Kyohei
Yamada[1], Motoki Sasakawa[1], Toshinobu Machida[1], Shin-Ichiro Nakaoka[1], Hideki Nara[1], Hiroshi
Tanimoto[1], Hitoshi Mukai[1], Yukio Yoshida[1], Shinji Morimoto[3], Shinya Takatsuji[4], Kazuhiro Tsuboi[4,5],
Yousuke Sawa[5], Hidekazu Matsueda[6], Kentaro Ishijima[5], Ryo Fujita[5], Daisuke Goto[7], Xin Lan[8,9],
Kenneth Schuldt[8,9], Michal Heliasz[10], Tobias Biermann[10], Lukasz Chmura[11,12], Jarsolaw Necki[11], Irène
Xueref-Remy[13]

[1]Earth System Division, National Institute for Environmental Studies, Tsukuba, Japan
[2]Graduate School of Agricultural and Life Sciences, The University of Tokyo, Tokyo, Japan
[3]Graduate School of Science, Tohoku University, Sendai, Japan
[4]Japan Meteorological Agency, Tokyo, Japan
[5]Department of Climate and Geochemistry Research, Meteorological Research Institute, Tsukuba, Japan
[6]Dokkyo University, Soka, Japan
[7]National Institute of Polar Research, Tachikawa, Japan
[8]Cooperative Institute for Research in Environmental Sciences, Boulder, USA
[9]Global Monitoring Laboratory, National Oceanic and Atmospheric Administration, Boulder, USA
[10]Centre for Environmental and Climate Science, Lund University, Lund, Sweden
[11]Faculty of Physics and Applied Computer Science, AGH University of Krakow, Krakow, Poland
[12]Institute of Meteorology and Water Management — National Research Institute, Warsaw, Poland
[13]Aix Marseille Univ, CNRS, Avignon Université, Institut de Recherche pour le Développement IRD, Institut Méditerranéen
de la Biodiversité et d'Ecologie marine et continentale IMBE, Aix-en-Provence, France

*Correspondence to*: Yosuke Niwa (niwa.yosuke@nies.go.jp)

## Abstract

An inverse study of atmospheric methane (CH₄) estimated regional and sectoral emission contributions to the unprecedented
surge of the atmospheric growth rate for 2020–2022. Three inverse analyses, which used only surface observations, surface
and aircraft observations, and satellite (GOSAT) observations, consistently suggested notable emission increases in the tropics
(15°S–10°N) (10–18 Tg CH₄ yr⁻¹) and in northern low-latitudes (10–35°N) (ca. 20 Tg CH₄ yr⁻¹), the latter of which likely
contributed to the growth rate surge from 2020. The emission increase in the northern low-latitudes is attributed to emissions
in South Asia (6–7 Tg CH₄ yr⁻¹) and northern Southeast Asia (5 Tg CH₄ yr⁻¹), which abruptly increased from 2019 to 2020,
and elevated emissions continued until 2022. Meanwhile, the tropical emission increase is dominated by tropical South
America (5–7 Tg CH₄ yr⁻¹) and central Africa (3–6 Tg CH₄ yr⁻¹), but they were continuously increasing before 2019.
Agreement was found in sectoral estimates in the tropics and northern low-latitudes, suggesting that biogenic emissions from



wetlands, agriculture, and waste are the largest contributors. High-precision surface and aircraft observations imposed constraints that were comparable to or 1.5 times stronger than GOSAT constraints on the flux estimates in South and Southeast Asia. Furthermore, a sensitivity inversion test suggested that the effect of the probable reduction of OH radicals in 2020 might

be limited in the Asian regions. These results highlight the importance of biogenic emissions in Asian regions for the persistent high growth rate observed during 2020–2022.

## 1 Introduction

Atmospheric methane ($CH_4$) is the second most important greenhouse gas (GHG) after carbon dioxide ($CO_2$). Sources of

atmospheric $CH_4$ exist at the Earth's surface, consisting of anthropogenic (60 %) and natural (40 %) emissions (Saunois et al., 2020). Meanwhile, the major sink, oxidation with OH radicals, exists in the atmosphere, which makes the lifetime of atmospheric $CH_4$ relatively short (ca. 9 yrs; Szopa et al., 2021). The source/sink imbalance determines the global growth rate of atmospheric $CH_4$. Excess emissions have increased atmospheric $CH_4$ by more than 250 % relative to the pre-industrial level (WMO, 2023). However, the growth rate of atmospheric $CH_4$ has not been steady; it decreased from the late 1980s, reaching

almost zero during 1999–2006, and then began to increase again starting in 2007 (Rigby et al., 2008). In 2020–2021, the growth rate rose sharply and reached the highest level (>15 ppb yr$^{-1}$) on record, followed by a continuously large growth rate of 13 ppb yr$^{-1}$ in 2022, before falling to the pre-surge level of 10 ppb yr$^{-1}$ in 2023 (Nisbet et al., 2023; Lan et al., 2024). Our understanding of these growth rate changes is insufficient, resulting in many controversial studies (e.g., Peng et al., 2022; Qu et al., 2022). It is imperative to evaluate a probable $CH_4$ emissions increase or OH sink decrease for 2020–2022. In particular,

$CH_4$ emission has recently drawn global attention because the mitigation effect of reducing $CH_4$ emissions on global warming occurs sooner than that of $CO_2$, and emission reduction is urgently required in the coming years through the target year of the Global Methane Pledge 2030.

        Previous studies (Peng et al., 2022; Qu et al., 2022; Stevenson et al., 2022; Feng et al., 2023) found that both increases in emissions and decreases in OH radicals contributed to the rise of atmospheric $CH_4$ in 2020. It is suggested that nitrogen

oxide (NOx) emissions largely decreased due to the lockdowns under the COVID-19 pandemic and consequently OH radicals decreased globally in that year (e.g., Miyazaki et al., 2021). Peng et al. (2022) and Stevenson et al. (2022) used chemical transport models to estimate that the global drop of OH radicals contributed about half of the atmospheric $CH_4$ increase in 2020. Meanwhile, although a significant contribution of OH radical was not denied, Qu et al. (2022) and Feng et al. (2023) estimated a larger contribution from an emissions increase. Particularly in 2021, when NOx emissions had recovered, a $CH_4$

emissions increase was likely the major driver of the $CH_4$ increase (Feng et al., 2023). In 2022, NOx and $CO_2$ emissions were reduced by the pandemic again, and the degree of decrease was even larger than it was in 2020 in China (Li et al., 2023). That was not the case globally (Liu et al., 2023), however, suggesting a continued contribution of emissions.





In this study, we investigated probable emission increases that induced the global atmospheric CH₄ surge for the entire period of high growth (i.e., 2020–2022). We took a so-called "top-down" approach, which derives information of emissions changes at the surface from observations of CH₄ mole fractions in the atmosphere. Specifically, we used an inversion method to quantitatively estimate spatiotemporal variations of surface CH₄ emissions with an atmospheric transport model and prescribed OH fields.

Several inversion analyses have been performed to investigate the recent growth in CH₄ emissions. Qu et al. (2022) and Feng et al. (2023) used column-averaged CH₄ data from the Japanese Greenhouse gases Observing SATellite (GOSAT: Kuze et al. 2009; Yokota et al. 2009), which is dedicated to observing CO₂ and CH₄, in their inversions and estimated that emissions from African wetlands had dominantly contributed the recent atmospheric CH₄ increase. The wetland emissions contributions were also suggested by the inversion of Peng et al. (2022) with in-situ and flask air sampling observations at ground-based stations, but the spatial coverage of the increased emissions ranged from the tropics to the Northern Hemisphere. The succeeding inversion of Lin et al. (2023) for 2020–2021 suggested contributions of wetland emissions in tropical Africa and Southeast Asia and attributed them to the La Niña event.

In fact, geospatial differences of increased CH₄ emissions in the previous inversions may have come from insufficient observational coverage and uncertainties. In-situ or flask air sampling measurements are precise, but their spatial coverages are limited. In particular, important CH₄ source regions at the low-latitudes (Asia, Africa and South America) remained poor in observations. Meanwhile, satellite data cover the globe relatively well, but they only provide column-averaged mole fractions in cloud-free areas. During winter seasons at high latitudes, satellite data are less available due to insufficient sunlight. Furthermore, they often have satellite-specific errors (Schepers et al., 2012). Specifically, different retrieval methods sometimes produce different features in mole fraction data products. Lin et al. (2023) used ground-based observations and data from two different GOSAT products. One is derived from the proxy method of the University of Leicester (Parker and Boesch, 2020), which is the same data that Qu et al. (2022) and Feng et al. (2023) used. The other is from the so-called full-physics retrieval method of the National Institute for Environmental Studies (NIES) (Yoshida et al., 2013). This type of multidirectional analysis is imperative to infer the observational uncertainties.

Inversions also have measurable uncertainties caused by the atmospheric transport model used as well as the inversion method (Saunois et al., 2020; Stavert et al., 2022). However, a limited number of transport models have been used in the previous studies. For example, Qu et al. (2022) and Feng et al. (2023) used the same transport model (GEOS-Chem), and Peng et al. (2022) and Lin et al. (2023) also used the same transport model (LMDZ-SACS), although Lin et al. (2023) tested different transport model configurations in their inversion analysis.

This study uses an inversion system based on a different transport model from those used in the previous studies. We here use the Nonhydrostatic Icosahedral Atmospheric Model (NICAM: Satoh et al., 2014)-based Inversion Simulation for Monitoring CH₄ (NISMON-CH₄). Using NISMON-CH₄, we estimate CH₄ emissions changes from in-situ and flask observations as well as GOSAT data. Moreover, for the in-situ and flask data, we use not only data obtained at the surface but also airborne data from various aircraft observations. Using these multiple observational platforms, we carefully evaluate the



reliability of the inversions. In addition, posterior flux errors are utilized to quantify observational impacts and the independence of estimated fluxes. Because our focus is on the emissions increase, the inversion analyses were performed with the climatological OH data, under the assumption that the OH field did not change from year to year. However, the effect of the probable OH reduction in 2020 was investigated by performing a sensitivity inversion test.

## 2 Method

### 2.1 NISMON-CH$_4$

The inverse analysis of NISMON-CH$_4$ uses a four-dimensional variational method (4D-Var) with the offline forward and adjoint tracer transport models of NICAM-based Transport Model (NICAM-TM: Niwa et al., 2011, 2017b). A similar inverse simulation for CO$_2$ using the same system (NISMON-CO$_2$) is described in detail by Niwa et al., (2017a, 2022). Because the CH$_4$ inverse analysis of this study adopts almost the same schemes used in Niwa et al. (2022), readers are encouraged to consult it for details. Unlike the conventional latitude-longitude grid system, NICAM has an icosahedral grid system, with hexagon- or pentagon-shaped grids. In this study, the model horizontal resolution is set at "glevel-5", which has a mean grid interval of approximately 223 km. The number of vertical layers is 40 with the top at approximately 45 km above sea level. Atmospheric transport fields to drive the offline NICAM-TM are given by a preliminary run of NICAM, with horizontal winds nudged to match Japanese 55-year Reanalysis data (JRA55: Kobayashi et al., 2015). The chemical reactions of CH$_4$ are calculated in NICAM-TM with prescribed chemical data that were used in the TransCom-CH$_4$ experiment (Patra et al., 2011): the tropospheric OH is derived but modified from the three-dimensional climatological fields of Spivakovsky et al., (2000), and the stratospheric reactions with Cl and O$^1$D are given by parameterized loss rates (Velders, 1995).

In NISMON-CH$_4$, although atmospheric transport is simulated on the icosahedral grids, fluxes are optimized on 1°×1° latitude-longitude grids through a grid conversion scheme (Niwa et al. 2022). For that flux optimization, a quasi-Newton method with the Broyden–Fletcher–Goldfarb–Shanno (BFGS) algorithm (Fujii, 2005; Niwa et al., 2017a) is used. Unlike the CO$_2$ inversion of Niwa et al. (2022), an external constraint is newly introduced in the cost function to avoid unrealistic negative values of CH$_4$ fluxes (or positive fluxes for soil uptakes). The details of this are described in Appendix A.

The inverse calculation period begins on 1 January 2015 with a three-dimensional CH$_4$ mole fraction field that is optimized by a previous inversion with surface observations (Saunois et al., 2020) and ends on 31 March 2023. To reduce errors induced by the initial mole fraction field (though, it is already optimized to a certain extent), the first 12 months (i.e., the year 2015) are disregarded in post-inversion analyses (i.e., it is the spin-up). Furthermore, the last three months (i.e., January–March 2023) are also disregarded because they might not be fully constrained by observations (spin-down). The 4D-Var method requires iterative calculations to optimize parameters. In the inversions described below, we confirmed that fluxes were well converged at the 200th iteration. Therefore, we commonly analyze flux data from the 200[th] iteration.



## 2.2 Flux model and prior flux data

In NISMON-CH4, the total net CH4 flux, $f_{CH_4}(x,t)$, consists of 10 sectoral fluxes (Table 1):

$$f_{CH_4}(x,t) = \sum_{i=1}^{5}\left(1 + \Delta\alpha_{anth,i}(x,t)\right)f_{anth,i}(x,t) + \left(1 + \Delta\alpha_{bb}(x,t)\right)f_{bb}(x,t) + \left(1 + \Delta\alpha_{nat}(x)\right)f_{nat}(x)$$

$$+\left(f_{rice}(x,t) + \Delta f_{rice}(x,t)\right) + \left(f_{wetl}(x,t) + \Delta f_{wetl}(x,t)\right) - \left(f_{soil}(x,t) + \Delta f_{soil}(x,t)\right), \qquad (1)$$

where $x$ and $t$ represent flux location and time, respectively. Optimizing parameters are described by $\Delta\alpha$ and $\Delta f$, which represent a modification factor and a flux deviation to each a priori sectoral flux, respectively. The first term in the right-hand

side denotes the sum of five anthropogenic (anth) emissions ($i$): coal mining (coa), oil/gas exploitation and use (ogs), landfill and waste (lfw), biofuels (bfl), and enteric fermentation and manure management (agr). Their prior fluxes $f_{anth,i}$ are taken from the annual mean data of Emissions Database for Global Atmospheric Research (EDGAR) version 6.0 (Crippa et al., 2021; Monforti-Ferrario et al., 2021). The temporal resolution of each $\Delta\alpha_{anth,i}$ is annual. The second term is biomass burning (bb) emissions, and its prior flux $f_{bb}$ is from the Global Fire Emission Database (GFED) v4.1s (van der Werf et al., 2017). The

temporal resolutions of $\Delta\alpha_{bb}$ as well as $f_{bb}$ are monthly. The third term represents natural (nat) emissions (the sums ocean, termite, and geological emissions); their prior fluxes $f_{nat}$ are derived from Weber et al. (2019), Ito (2023), and Etiope et al. (2019), respectively (the geological emissions are scaled so that the global total is 23 Tg according to Canadell et al. (2021)). Because their interannual variations are highly uncertain and their contributions are minor compared to the other fluxes, $f_{nat}$ and $\Delta\alpha_{nat}$ are set to be temporally constant throughout the analysis period. The latter three terms are monthly emissions of

rice cultivation, wetland, and soil uptakes, and their prior fluxes, $f_{rice}$, $f_{wetl}$, and $f_{soil}$, are given by the terrestrial biosphere model Vegetation Integrative SImulator for Trace gases (VISIT: Ito and Inatomi, 2012). This study uses the fluxes that are calculated with the scheme of Cao et al. (1996) for $f_{rice}$ and $f_{wetl}$. As of the start of this study, the data from EDGAR ($f_{anth,i}$) and VISIT ($f_{rice}$, $f_{wetl}$ and $f_{soil}$) were available through 2018 and 2020, respectively. For the later years, we used data from the final year they were available. Consequently, during the period of 2020–2022, which is the focus of this study, prior fluxes

other than the biomass burnings do not have interannual variations. Therefore, in the inversions, interannual variations of estimated emissions are mostly derived from observations.

As shown in Eq. (1), the flux optimization parameters are constructed by the mixture of the scaling factors and the flux deviations. In fact, they are applied to spatially small-scale ($f_{anth,i}$ and $f_{bb}$) or minor ($f_{nat}$) fluxes and to those with relatively broad-scale variations ($f_{rice}$, $f_{wetl}$, and $f_{soil}$), respectively. The scaling factor only modifies flux magnitudes but not

distributions because the inversion may not be able to modify small-scale distributions reliably due to the nature of atmospheric mixing.

In the inversion, the prior errors of the scaling factors are set at 50 % for $\Delta\alpha_{anth,i}$ and 100 % for $\Delta\alpha_{bb}$ and $\Delta\alpha_{nat}$ without error covariances. The prior errors and error covariances of $\Delta f_{rice}$, $\Delta f_{wetl}$, and $\Delta f_{soil}$ are derived from ensembles. Each ensemble is calculated from a 120-year-long simulation (1901–2020) of VISIT, in which data in each year are considered as

one member. A similar method is used in NISMON-CO2 and is detailed in Niwa et al. (2022). Those prior errors and



covariances are combined to construct a prior error covariance matrix, **B**, with which the cost function of the inversion is defined.

**Table 1** Categorization of CH$_4$ fluxes

| Sector | Notation | Prior flux data | Merged sector |
|---|---|---|---|
| wetland | wetl | VISIT (Ito and Inatomi, 2012; Cao et al., 1996) | Wetland |
| rice cultivation | rc | VISIT (Ito and Inatomi, 2012; Cao et al., 1996) | Agriculture & waste |
| agriculture other than rice cultivation | agr | EDGAR ver.6 (Crippa et al., 2021; Monforti-Ferrario et al., 2021) | |
| landfills and waste | lfw | EDGAR ver.6 (Crippa et al., 2021; Monforti-Ferrario et al., 2021) | |
| coal mining | coa | EDGAR ver.6 (Crippa et al., 2021; Monforti-Ferrario et al., 2021) | Fossil fuel |
| oil/gas exploitation and use | ogs | EDGAR ver.6 (Crippa et al., 2021; Monforti-Ferrario et al., 2021) | |
| biofuel | bfl | EDGAR ver.6 (Crippa et al., 2021; Monforti-Ferrario et al., 2021) | Biomass burning |
| biomass burning | bb | GFED v4.1s (van der Werf et al., 2017) | |
| ocean, termite, and geological emissions | nat | Weber et al. (2019), Ito and Inatomi (2012), and Etiope et al. (2019) | N/A |
| soil uptake | soil | VISIT (Ito and Inatomi, 2012) | |


**2.3 Observations**

**2.3.1 Surface and aircraft data**

In this study, we performed two inversions with in-situ and flask observations (listed in Supplementary materials 1 and 2): one uses only surface observations (SURF),  and the other uses both surface and aircraft observations (SURF+AIR). These in-situ
and flask data were obtained from version 6.0 of ObsPack GLOBALVIEWplus (Schuldt et al., 2023a). In addition, data from version 6.0 data of ObsPack Near Real Time (NRT) (Schuldt et al., 2023b) were also used for 2023 (spin-down period). Furthermore, we used additional data from NIES and collaborative networks: flask air sampling observations at Asian stations (Tohjima et al., 2002, 2014; Nomura et al., 2017, 2021; Okamoto et al., 2018); flask and in-situ continuous observations on voluntary observing ships (VOS) in the Pacific, Oceania, and around Southeast Asia (Terao et al., 2011; Nara et al., 2017);
and flask observations by aircraft over Siberia (Sasakawa et al., 2017). We also used in-situ continuous data from the Japan-Russia Siberian Tall Tower Inland Observation Network (JR-STATION) operated by NIES (Sasakawa et al., 2010) and flask observations from the aircraft programs of the Comprehensive Observation Network for TRace gases by AIrLiner



(CONTRAIL: Machida et al., 2008; Matsueda et al., 2015; Sawa et al., 2015; Umezawa et al., 2012) and Tohoku University (Umezawa et al., 2014).

185       The ObsPack datasets provide $CH_4$ mole fractions on the WMO $CH_4$ X2004A scale (Dlugokencky et al., 2005), while the NIES and CONTRAIL data are provided on the NIES94 $CH_4$ standard scale, which is approximately 5 ppb higher than the WMO scale (Machida et al., 2023). The aircraft data from Tohoku University are based on the TU-1987 scale, which is deemed to be comparable to the WMO scale (Fujita et al., 2018). In this study, we commonly use the $CH_4$ observations on the WMO scale; we modified the NIES-94 scale data to the WMO scale by using the linear relationship reported in Tsuboi et al. (2017).

190       Figure 1a shows the geographic locations of surface observations used in SURF. Here, we use almost all available data from the ObsPack datasets for 2015–2023, but we use only the highest altitude data for tower sites that provide data for multiple altitudes. The ship observations that cover the northern Pacific, Asia and Oceania regions are from the NIES VOS program. Note that data at each site in Fig. 1a are not always available for the whole period.

      The locations of the aircraft data used in SURF+AIR are depicted in Fig. 1b. The network covers various regions by 195 many campaign flights, such as by ACT America (Wei et al., 2021; Davis, et al., 2018). The National Oceanic and Atmospheric Administration (NOAA) and Japan Meteorological Agency (JMA) operate regular aircraft observations at fixed areas over North America (Sweeney et al., 2015) and the western North Pacific (Tsuboi et al., 2013; Niwa et al., 2014), respectively. Observations using commercial airliner are also regularly operated by CONTRAIL and In-service Aircraft for a Global Observing System (IAGOS: Schuck et al., 2012; Petzold et al., 2015), but their flight routes change frequently. Nevertheless, 200 CONTRAIL has continuously provided data from around Asian regions during the analysis period. Aircraft often enters the lower stratosphere (LS), and this is especially true for commercial flights because they fly at higher altitudes (~10 km). Variations of $CH_4$ in LS are largely affected by the stratospheric circulation, and consequently, their seasonal patterns differ largely from those in the upper troposphere (UT) (Sawa et al., 2015). Therefore, we use only tropospheric data for the aircraft observations. For the data selection, we use a potential vorticity (PV) criterion of 2 PVU (1 PVU = $10^{-6}$ $m^2$ $s^{-1}$ K $kg^{-1}$), which 205 is from the NICAM simulation; observations where absolute values of simulated PV were larger than 2PVU were excluded.

      As shown in Fig. 1a, the observation network is dense in Europe and North America, while regions such as South America and Africa have fewer observations. Furthermore, in-situ and flask data have different temporal resolutions; typically, they are hourly and (bi-)weekly, respectively, at ground-based stations. Such data inhomogeneity due to different measurement methodologies also exists in the aircraft observations. In the inversions, we therefore introduce observational weighting, in 210 which a diagonal element of the observation-model mismatch error covariance matrix (here assumed as a diagonal matrix), $\boldsymbol{R}$, is defined as

$$R_{ii} = (\beta r_i)^2 N_i, \qquad (2)$$

where $N_i$ denotes the number of observations within a certain spatiotemporal range of the $i$th observation. In this study, the range is set at one week, a 1000-km horizontal diameter circle, and a 1-km vertical depth. $r_i$ represents the standard deviation 215 of mole fraction variations of the $i$th observation (from 1 week before to 1 week after at the same location), which is derived from the NICAM simulation with prior fluxes or observations (the larger one is chosen). $\beta$ is a tuning parameter to balance the





weight with the prior estimate and is here set at 0.5. Both the SURF and SURF+AIR inversions use Eq. (2) with the same configuration. Through the Eq. (2), although in-situ and flask observations are high-precision and their uncertainties are within a few ppb, an observation-model mismatch error (the square root of $R_{ii}$) reaches over 100 ppb where observations are densely existing.

### 2.3.2 GOSAT

We also used GOSAT column-averaged dry-air mole fraction (XCH4) data from the full-physics retrieval of NIES (Yoshida et al., 2011, 2013); they are provided as a GOSAT Level 2 (L2) CH4 product from short wavelength infrared (SWIR) spectral data observed by GOSAT Thermal And Near infrared Sensor for carbon Observation-Fourier Transform Spectrometer (TANSO-FTS). The version 2.95/96 for General Users was used. Compared to the proxy method (Parker and Boesch, 2020), the full-physics method uses a radiative transfer model that considers multiple scattering by aerosols and clouds explicitly. Although the number of observations that are well retrieved is smaller, the full-physics method is less sensitive to prior model $CO_2$ data than the proxy method (Schepers et al., 2012). As shown in Fig. 1c, GOSAT data cover the globe well between 60°S and 60°N, but the latitudinal range of data available changes seasonally because of sunlight (data higher than approximately 45°N are not available during northern winter). In the inversion, we used all available data including sunglint condition data; they were all corrected in advance by the bias evaluation of Inoue et al. (2016) (NIES GOSAT Project, 2023).

The GOSAT inversion of NISMON-CH4 has the satellite-specific observational operator between the simulated and observed mole fractions. A simulated dry-air column averaged mole fraction corresponding to a GOSAT observation, $X^{\mathrm{sim}}$, is calculated from a vertical profile of simulated dry-air mole fractions, $c^{\mathrm{sim}}$, as

$$X^{\mathrm{sim}} = w^T A (G c^{\mathrm{sim}} - c^{\mathrm{pri}}) + w^T c^{\mathrm{pri}}, \qquad (3)$$

where $w$ is a weighting vector based on pressures. The matrices of $A$ and $G$ are an averaging kernel matrix and a remapping matrix from the model vertical layers to the GOSAT retrieval layers, respectively, and $c^{\mathrm{pri}}$ is a vertical profile vector of CH4 dry-air mole fractions that is used as *a priori* in the retrieval. $A$ and $c^{\mathrm{pri}}$ are provided with the GOSAT-L2-SWIR product.

Because GOSAT has relatively homogeneous data in space and time compared to the in-situ and flask observations, we do not employ the observational weighting of Eq. (2) in the GOSAT inversion. Instead, we commonly set the observation-model mismatch error (the square root of a diagonal element of the error covariance) as 20 ppb for every observation. In fact, it is larger than the probable error of the GOSAT data (NIES GOSAT Project, 2023). This error inflation is intended to implicitly consider error correlations among nearby observations. As is done for the SURF and SURF+AIR inversions, off-diagonal elements of the error covariance matrix are set at zero.

Although the GOSAT data are corrected separately for land and ocean (NIES GOSAT Project, 2023), it is known that some spatial (e.g., latitudinal) biases still exist in the satellite data, which is often corrected before inversion by referring to an independent inversion with high-precision in-situ/flask observations (Bergamaschi et al., 2007, Meirink et al., 2008). In this study, we did not apply this type of bias correction for the GOSAT data so that the SURF/SURF+AIR and the GOSAT inversions remained independent. The probable spatial bias of the GOSAT data is discussed in Appendix B.




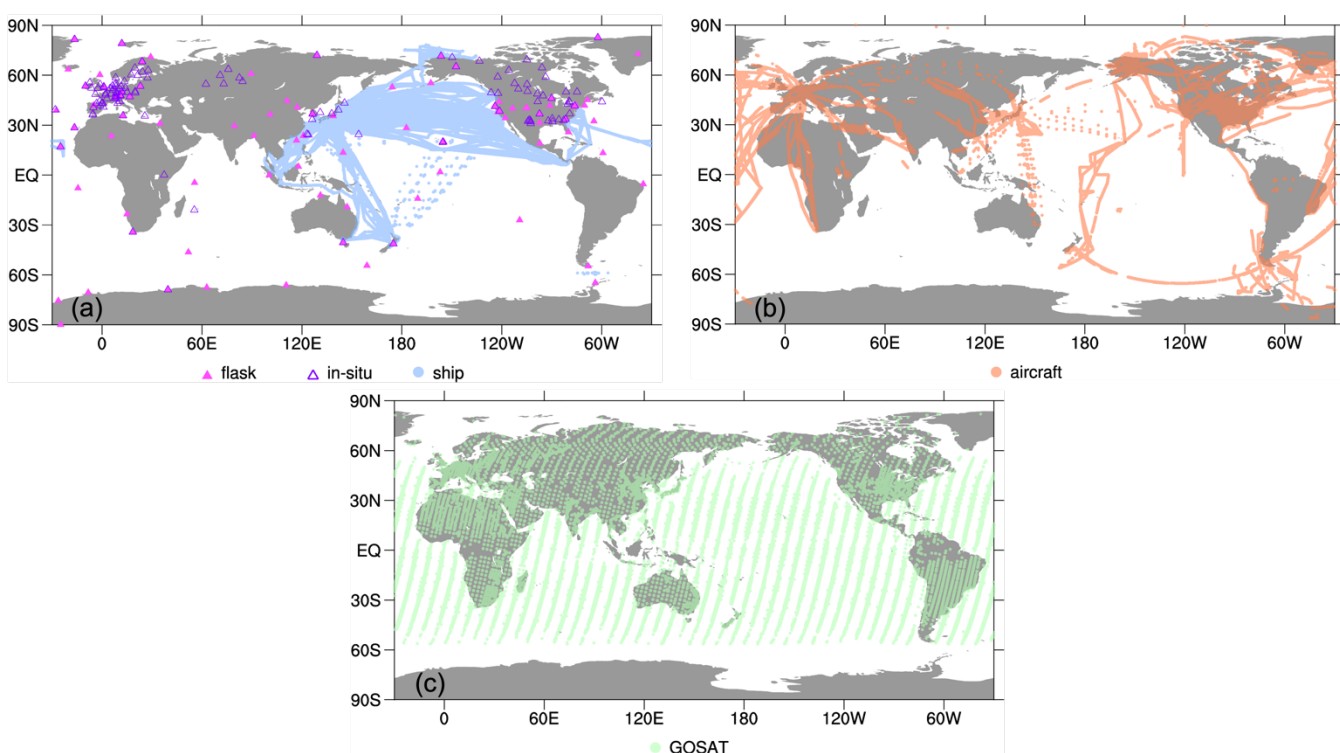

**Figure 1** Locations of the observations used in the inverse analyses. In-situ (open triangles) and flask air (closed triangles) measurements at surface stations and ship (light blue dots) data for January 2015–March 2023 are shown in (a) and those for aircraft (orange) are shown in

(b). The GOSAT data (light green) shown in (c) are obtained during 2020, which has a similar pattern to the other years.

## 2.4 Posterior errors

Observational constraints in the inversions are quantified by using posterior errors. Posterior errors are derived from diagonal elements of the posterior error covariance matrix, $P$, which can be written with the error covariance matrices $B$ and $R$ and with

the linear model operator matrix of atmospheric transport $M$ (including the observational operator and the flux model) as

$$P = (B^{-1} + M^T R^{-1} M)^{-1}. \qquad (4)$$

Because the matrix size of Eq. (4) is extremely large in the 4D-Var method, which optimizes flux parameters at each grid (1°×1° in this study), it is impossible to analytically calculate Eq. (4). Therefore, we use the approximation method of Niwa and Fujii (2020) to estimate each element of $P$, which uses the BFGS formula with vector pairs generated from ensemble

calculations and orthogonalization. This method can estimate $P$ accurately, not only for diagonal but also for off-diagonal elements for a linear problem. In fact, the inverse problem in this study is nonlinear because an additional constraint is introduced to avoid negative values (Appendix A). Therefore, to estimate $P$, we omit the nonlinear additional constraint.



Furthermore, we calculate $P$ only for the year 2020 due to the high computational demand. In the three inversion cases, we performed 50 iterations with 120 ensemble members. From the 6000 vector pairs generated, we obtained approximately 900

conjugate vector pairs by orthogonalization. Those conjugate vector pairs were used to estimate $P$ with the BFGS formula. Then, we applied spatiotemporal aggregation as was done for the posterior fluxes:

$$P^{agr} = DPD^T, \qquad (5)$$

where $D$ and $P^{agr}$ are an aggregation operator matrix and an aggregated posterior error covariance matrix, respectively. The square root of the $i$th diagonal element of $P^{agr}$ is the posterior error of the $i$th aggregated flux parameter, $\sigma_i^{pos}$. Its error reduction

ratio from a corresponding prior error $\sigma_i^{pri}$ defined as

$$e_i = \frac{\left(\sigma_i^{pri} - \sigma_i^{pos}\right)}{\sigma_i^{pri}} \times 100 \qquad (6)$$

can be used to quantify the strength of the observational constraint imposed on the $i$th flux parameter; a larger $e_i$ means a stronger constraint by observations. Furthermore, an off-diagonal element of $P^{agr}$ derives an error correlation between two flux parameters, which could indicate how independently a flux parameter is optimized in inversion.


## 2.5 Sensitivity tests

To investigate the effects of the probable OH reduction due to the pandemic in 2020, we performed an extra inversion analysis with a modified OH field. The methods and results of this analysis are described in Appendix C. Furthermore, we also performed inversion analyses with different datasets of in-situ and flask observations as well as GOSAT retrieval. The details

of the observations and the results of these inversions are described in Appendix D.

## 3 Results

### 3.1 Evaluation of posterior mole fractions

Before evaluating the posterior fluxes, we evaluated the consistency of posterior atmospheric $CH_4$ mole fractions with observations to assess the validity of our inversions. For the evaluation, we calculated correlations and root-mean-square

differences (RMSDs) between the model and the observations for the northern high- and low-latitudes (35–90°N, 10–35°N), the tropics (15°S–10°N), and the southern latitude (90–15°S).

For the reference observations, we used flask air sampling observations from surface sites (including ships) and aircraft, which are part of the observations used in the inversions. The in-situ data were not used to avoid excessive weights of those data on the evaluation. Furthermore, the flask-sampling data at Comilla, Bangladesh, were excluded because the observed

mole fractions at Comilla were known to have extremely large variations (Nomura et al., 2021), and the observation-model mismatch would induce a large weight of South Asia in the statistics (note that the Comilla data were used in the inverse





analysis because Eq. (2) made observational weights flatter). For the aircraft data, we used only data from over 3 km altitude to represent the free troposphere.

    In addition, we also used the same GOSAT data used in the inversion. Before calculating the statistics, we subtracted

the averages for 2016–2019 from the modelled and observed mole fractions, respectively, for each latitudinal band, which excludes temporally- and spatially- varying biases of the GOSAT data that may still exist after the globally uniform bias correction. However, as shown in Appendix B, we found notable systematic differences between the SURF (or SURF+AIR) and GOSAT inversions. Therefore, in this study, we only discuss differences of fluxes or atmospheric $CH_4$ from their averages for the former period (2016–2019) by latitude or region. This could cancel the spatial biases of GOSAT and enabled us to

focus on the increases of fluxes or mole fractions for 2020–2022, which is the target of our study.

    Figure 2 shows the calculated correlations and RMSDs between the simulated and observed atmospheric $CH_4$. In most cases, the posterior correlations and RMSDs are larger and smaller, respectively, than those of priors, indicating that the inversions appropriately incorporated the observations in their flux optimizations. In particular, the larger correlations and smaller RMSDs with independent observations (i.e., those not used in the inversion) suggest that posterior $CH_4$ fluxes have

improved atmospheric $CH_4$ fields relative to the prior ones. The GOSAT inversion shows little improvement of or even worse correlations and RMSDs when evaluated against the surface observations. However, it has clearly better correlations and RMSDs against the aircraft observations than the prior ones, which are even better than the SURF inversion. This is attributable to the fact that both the aircraft observations and the column-averaged observations of GOSAT better represent well-mixed conditions in the free troposphere than the surface observations; therefore, their footprints might be more similar to each other

than to those of the surface observations.

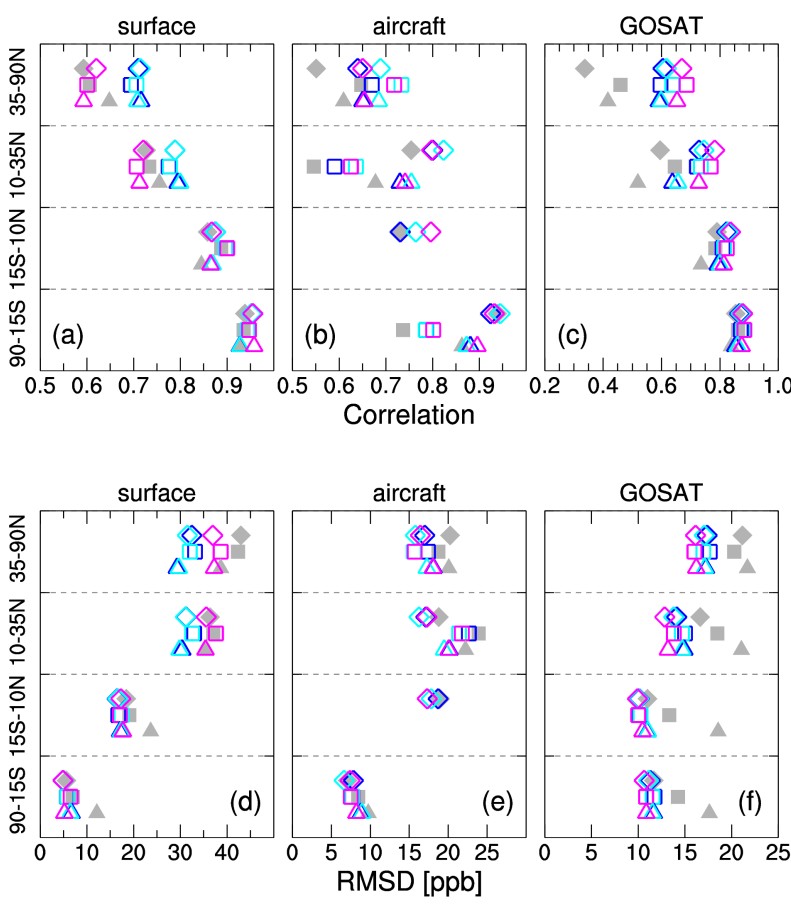

**Figure 2** Correlations (upper panels) and root-mean-square-differences (RMSDs) (lower panels) between observations and prior or posterior mole fractions of atmospheric CH$_4$ in northern high-latitudes (35–90°N), northern low-latitudes (10–35°N), tropics (15°S–10°N), and southern latitudes (90–15°S) for 2020 (diamonds), 2021 (squares) and 2022 (triangles). The correlations and RMSDs are calculated for surface (with Comilla excluded) (a, d) and aircraft (> 3km) (b, e) flask observations, as well as for the GOSAT data (c, f). The prior and posterior mole fractions are derived from atmospheric simulations of NICAM-TM with the prior and posterior (SURF, SURF+AIR, and GOSAT) fluxes, respectively. Before the calculations for correlations and RMSDs, the average for 2016–2019 is subtracted from the observations and simulated mole fractions for 2020–2022 for each observational type.

## 3.2 Global features

Figure 3 shows the spatial patterns of the posterior total net CH$_4$ emissions for the pre-growth period of 2016–2019 as well as the patterns of the (posterior – prior) differences. In general, the estimated spatial patterns are consistent in the three inversions, but the differences have different features between the SURF or SURF+AIR inversion and the GOSAT inversion (Figs. 3b–d). Specifically, the tropical (e.g., the central Africa and the tropical South America) fluxes are noticeably larger than the prior



fluxes for the GOSAT inversion. Meanwhile, those of SURF and SURF+AIR are rather consistent with the prior fluxes. Therefore, the GOSAT inversion has systematically larger emissions in the tropics than the other two inversions.

Despite such differences among the posterior fluxes, spatial patterns of the CH₄ emission increases from 2016–2019 to 2020–2022 are consistent in the three inversions (Fig. 4). Hereafter, we refer to a difference of CH₄ emissions from the

mean for 2016–2019 as $\Delta f CH_4$. In Fig. 4, the area with notable positive $\Delta f CH_4$ values ranges from the tropics to the northern high-latitudes. The increase in the northern low-latitudes (10–35°N) is particularly noteworthy; the three inversions consistently estimated that northern South Asia (Bangladesh and northern India) and the Indochina Peninsula were major contributors to the increase. Meanwhile, in the northern high-latitudes (35–90°N) and topics (15°S–10°N), areas with a large emissions increase are also consistently estimated, but their magnitudes differ, especially between the SURF or SURF+AIR

inversion and the GOSAT inversion. For example, emissions in the northern North America and Sahel regions are estimated as smaller and larger, respectively, in the GOSAT inversion.

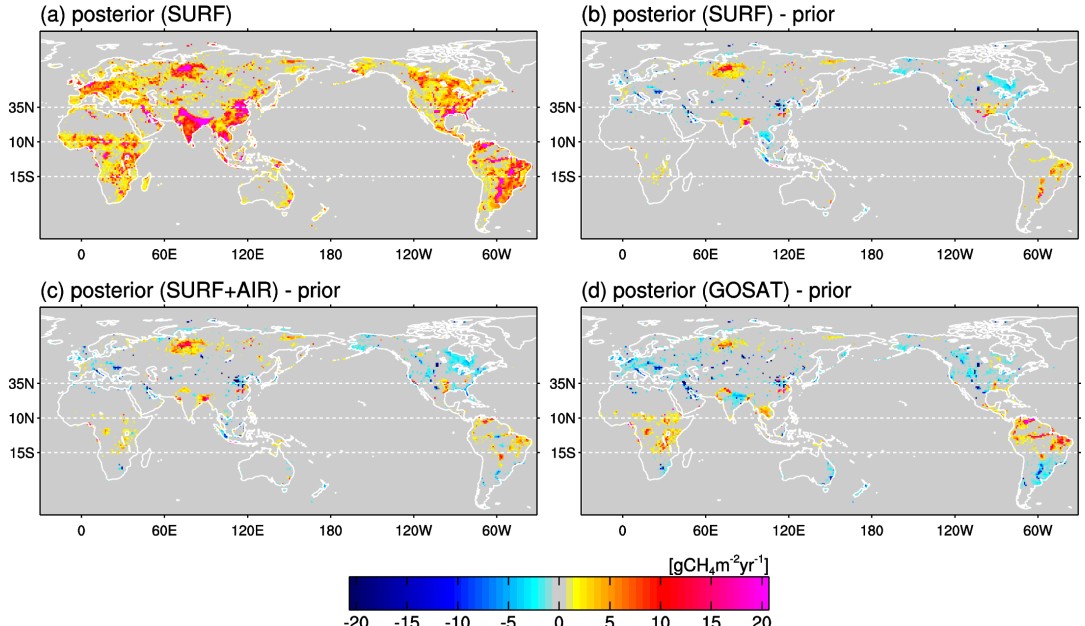

**Figure 3** Spatial pattern of posterior total net CH₄ emissions by the SURF inversion averaged for 2016–2019 (a) and the (posterior – prior) difference pattern (b). Also shown are the other (posterior – prior) emissions patterns for the SURF+AIR (c) and GOSAT (d) inversions.




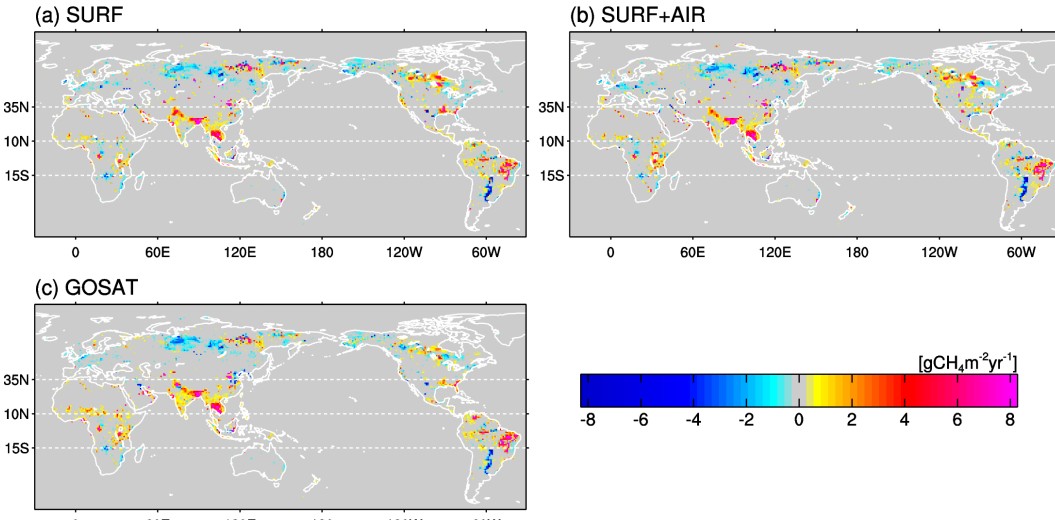

**Figure 4** Spatial patterns of the total net CH₄ emissions increase (ΔfCH₄) from 2016–2019 to 2020–2022 in the SURF (a), SURF+AIR (b), and GOSAT (c) inversions.

Year-to-year variations of the global total net CH₄ emissions are also consistently estimated by the inversions for 2016–2022 (Fig. 5a). Their temporal patterns are similar to those of the global growth rate observed in the marine boundary layer sites of NOAA (Lan et al., 2024); in 2020, the global total net emissions abruptly increased by approximately 30 Tg CH₄ yr⁻¹, followed by a similar magnitude or even greater increased emissions in 2021. In 2022, the emissions decreased but remained greater than the pre-2020 level. In 2021, when the inversions showed the highest emissions, the estimates differ largely ranging from 592 Tg CH₄ yr⁻¹ (SURF) to 603 Tg CH₄ yr⁻¹ (GOSAT). Also, in the three different latitudinal bands, ΔfCH₄ changes are consistently estimated (Figs. 5b–e). The increase in the northern low-latitudes is noteworthy; it has a sharp rise from 2019 to 2020, and large emissions continue through 2022, when the magnitudes (ca. 20 Tg CH₄ yr⁻¹) are consistent among the inversions. Also, in the tropics, the inversions consistently show increases of 10–18 Tg CH₄ yr⁻¹ in 2020–2022; however, there are gradual increases from 2016, but they do not largely contribute to the global surge in 2020. Meanwhile, in the northern high-latitudes, SURF and SURF+AIR estimated a marginal increase that has a peak of 9 Tg CH₄ yr⁻¹ in 2021, while GOSAT estimated smaller increases of 2 Tg CH₄ yr⁻¹. In the southern latitudes, all the inversions do not show any notable change during 2016–2022.



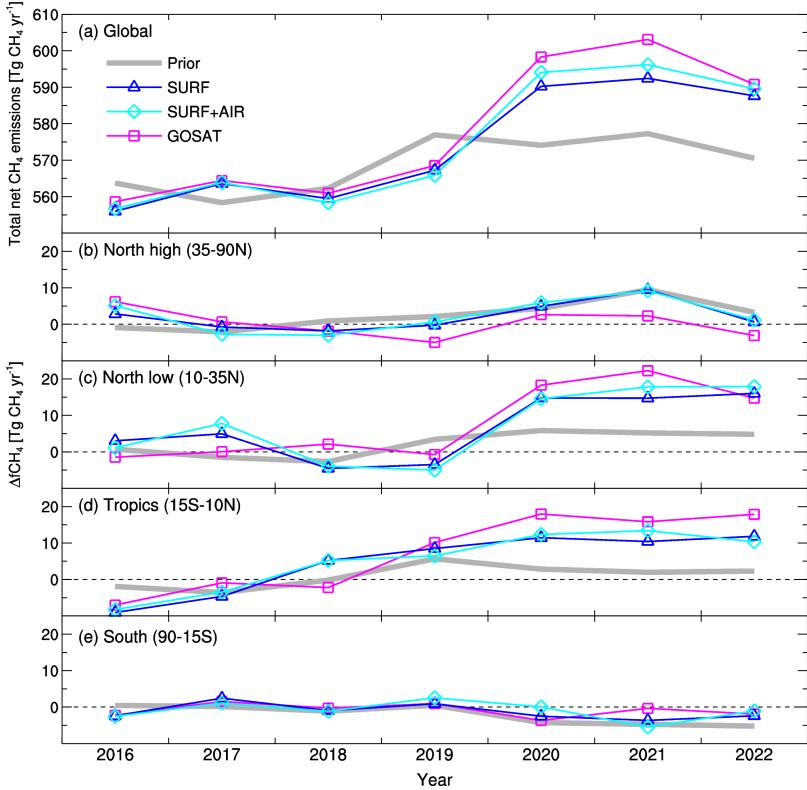

**Figure 5** Year-to-year variations of total net CH$_4$ emissions integrated globally (a) and those of ΔfCH$_4$ integrated in each latitudinal band: northern high-latitudes (35–90°N) (b), northern low-latitudes (10–35°N) (c), tropics (15°S–10°N), and southern latitudes (90–15°S) (e).

## 3.2 Regional features

Figure 6 shows a further regional breakdown of the ΔfCH$_4$ changes. Figure 6 shows that, even for these smaller regions, the three different inversions consistently show temporal variations of ΔfCH$_4$ in general. In particular, the consistency between the SURF and SURF+AIR inversions and the GOSAT inversion is noteworthy, because the in-situ or flask observation and the GOSAT data are completely independent from each other.

Specifically, in northern Southeast Asia and South Asia, both of which are located in the northern low-latitudes (10–35°N), the abrupt increase of ΔfCH$_4$ by 5 Tg CH$_4$ yr$^{-1}$ or more in 2020 and its continuation until 2022 is consistently estimated by all the inversions. This suggests that these two regions are dominant contributors to the recent surge of atmospheric CH$_4$. Meanwhile, in West Asia, the drop in 2019 is notable and is also estimated consistently by the inversions. In Northern Africa, the SURF+AIR and GOSAT inversions estimated marginal increases of CH$_4$ (ca. 3 Tg CH$_4$) in 2020, while SURF estimated a more moderate and gradual increase until 2022. Meanwhile, the inversions show ΔfCH$_4$ increases up to 4 Tg CH$_4$ yr$^{-1}$ in East Asia for 2020–2021, although the GOSAT inversion shows larger interannual variations during the analysis period. These flux changes may also have contributed to the surge of atmospheric CH$_4$ in 2020 to some extent.





Two regions, Central Africa and tropical South America, contribute to the gradual increases of $\Delta fCH_4$ in the tropics (Fig. 5d). Meanwhile, $\Delta fCH_4$ also increased in southern Southeast Asia, but only during the middle of the analysis period (2019–2021). Although there are some discrepancies, these features of tropical emissions are commonly seen in the three inversions. The relatively large increase of $\Delta fCH_4$ in Central Africa, which is only estimated by the GOSAT inversion, might

have contributed to the surge of atmospheric $CH_4$, but only for 2020.

In the northern high-latitude areas, moderate increases of up to 3 Tg $CH_4$ yr$^{-1}$ are estimated in the west part of Northern Eurasia for 2020 and in boreal North America for 2020–2021 by SURF and SURF+AIR, but these increases are not clearly seen in GOSAT. The east part of Northern Eurasia shows a notable peak in 2021, with magnitudes of 5 Tg $CH_4$ yr$^{-1}$ and 2 Tg $CH_4$ yr$^{-1}$ in SURF or SURF+AIR and GOSAT, respectively. For Europe and the western part of northern Eurasia, the

inversions suggested that $CH_4$ emissions have not significantly contributed to the increase of atmospheric $CH_4$ during 2020–2022.

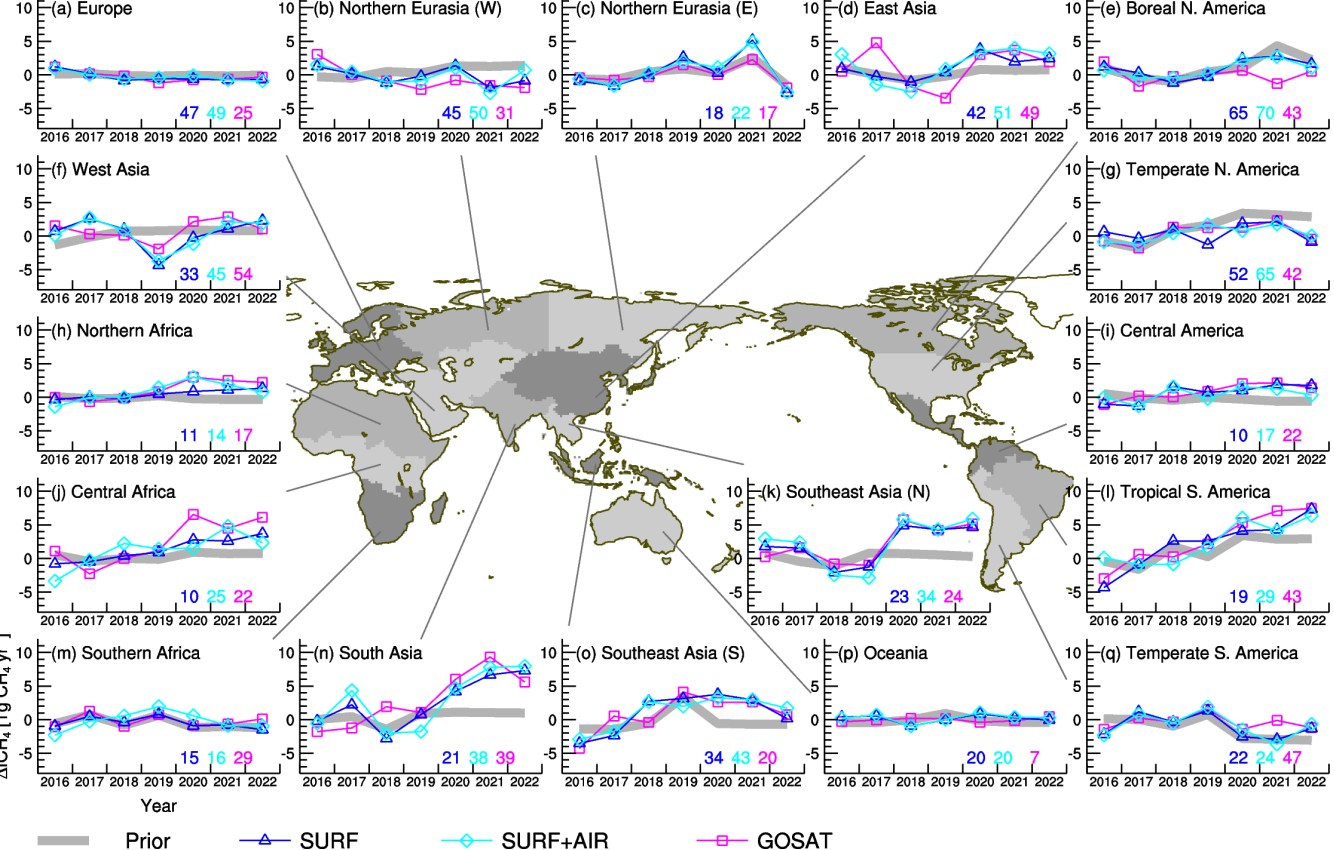

**Figure 6** Year-to-year variations of regional $\Delta fCH_4$. Emissions are integrated within each geographical region defined in the centered map, which is the same as that used in Canadell et al. (2021), except that northern Eurasia and Southeast Asia are further divided by west/east and



north/south. The numbers in each panel denote the uncertainty reduction ratio of the annual flux for 2020 in each region, with the color corresponding to that of $\Delta f CH_4$.


The regional $\Delta f CH_4$ for 2020–2022 estimated by the inversions is summarized in Fig. 7. In total, the SURF and GOSAT inversions estimated emission increases of 29 Tg CH$_4$ yr$^{-1}$ and 34 Tg CH$_4$ yr$^{-1}$, respectively. The SURF+AIR inversion estimated an intermediate value of 32 Tg CH$_4$ yr$^{-1}$. Three regions—South Asia, northern Southeast Asia, and tropical South America—are commonly presented as major contributors; their estimated emission increases are 6–7 Tg CH$_4$ yr$^{-1}$, 5 Tg CH$_4$ yr$^{-1}$, and 5–7 Tg CH$_4$ yr$^{-1}$, respectively. The SURF and SURF+AIR inversions suggested the northern regions (Northern Eurasia (E) and Boreal North America) as marginal contributors, but their contributions estimated by the GOSAT inversion are much smaller. Meanwhile, estimated contributions from the African regions (Northern Africa and Central Africa) are larger for GOSAT (3 and 6 Tg CH$_4$ yr$^{-1}$, respectively) than for the other two (1–2 and 3 Tg CH$_4$ yr$^{-1}$, respectively). Interestingly, all the inversions agree with each other in that the five Asian regions contributed by approximately 60 % of the global $\Delta f CH_4$ increase.

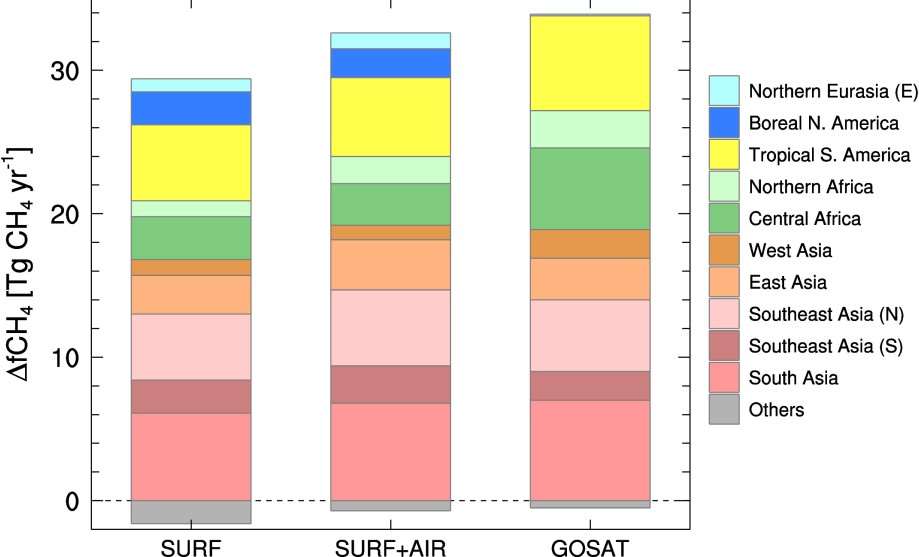

**Figure 7** Cumulative bar chart of $\Delta f CH_4$ for 2020–2022 estimated by the three inversions (SURF, SURF+AIR, and GOSAT). Contributions from regions where notable emissions changes occurred (Fig. 6) are noted by the colored areas and the others are aggregated into the gray "Others" category. If a mean $\Delta f CH_4$ is positive (negative), it is accumulated upward over (downward under) the dashed zero line.

In fact, these estimated regional $\Delta f CH_4$ increases may have non-negligible uncertainties, a major cause of which is the sparseness of observations. Uncertainties caused by insufficient observations can be inferred from the uncertainty reduction ratios depicted in each regional panel of Fig. 6 (bottom-right numbers). In the northern high-latitude areas, the constraints of





SURF and SURF+AIR are stronger than those of GOSAT, which is attributable to the dense in-situ and flask observation network in the area (Figs. 1a and b) and also to the limitations on GOSAT observations during winter. Meanwhile, the constraints of SURF and SURF+AIR are weaker in the lower latitudes, which is attributable to the decreased availability of observations. Nevertheless, the Asian regions had relatively strong constraints from the in-situ and flask observations, especially in the case of SURF+AIR. These strong constraints are attributed to the ground-based stations and ship observations

operated by NIES (Appendix D). Furthermore, these regions are further constrained by aircraft data in the upper-air, most of which are contributed by CONTRAIL and JMA aircraft on the downwind side of the continent. Consequently, the observational constraints of SURF+AIR in the Asian regions are comparable to or larger than those of GOSAT. Specifically in southern Southeast Asia, where cloud cover is dense and active convection effectively lifts flux signals up to the upper-air (Niwa et al., 2012, 2014, 2021), the superiority of SURF+AIR to GOSAT is pronounced. These stronger constraints give a

higher confidence about the temporal changes of the estimated $CH_4$ emissions. Regional $CH_4$ emission changes that were consistently estimated by different inversions with a large range of constraints might be derived from a large scale observational information, not necessarily from regionally available observations.

Figure 8 shows error correlations of the regionally aggregated posterior fluxes, which are derived from the off-diagonal elements of the posterior error covariance matrix of Eq. (5). If two regions are anti-correlated (which is more or less

true in most cases), estimated flux values might be compensating for each other (i.e., the fluxes are not independently estimated). In general, the three inversions have a similar anti-correlation pattern, indicating that that feature is mostly determined by factors other than the observations used (e.g., atmospheric transport or prior flux errors and error correlations). On a broader scale, the inversions commonly have notable error correlations among the northern high-latitude areas (A–E in Fig. 8), among South America (G and H), and between the tropical South America (G) and African regions (I–K). South Asia

and northern Southeast Asia, which are the largest contributors to the 2020–2022 atmospheric $CH_4$ growth, are anti-correlated with each other, especially when observational constraints are strong (i.e., with SURF+AIR and GOSAT), indicating that the separation of these two regions has uncertainties. However, their anti-correlations with other regions are minor, indicating that the sum of the two is independently estimated by the inversions. Therefore, it is likely that either or both of the two regions contributed to the 2020–2022 atmospheric $CH_4$ growth. Interestingly, for other areas, fluxes where a dense observational

network is available are not always independently estimated. For instance, the error correlation between boreal and temperate North America is notably large, though they have quite dense observational networks (Fig. 1).




**Figure 8** Error correlations of the regionally aggregated posterior fluxes for 2020 (annual mean) in the SURF (a), SURF+AIR (b), and GOSAT (c) inversions. A negative value indicates that estimated fluxes are anti-correlated with each other. The dotted black lines group
regions in a broader scale (e.g., Asia for L–P, Eurasia for D and E).

### 3.3 Sectoral contributions

Although our inversion system does not currently incorporate isotope data to separately evaluate sectoral contributions (Lan et al., 2021; Chandra et al., 2024), we optimized CH$_4$ emissions by sector with the expectation that spatial and temporal
variations of observations could provide information about sectoral contributions to some extent. If different sectors do not overlap with each other in space and time, they might be optimized independently. However, it would largely depend on prior emissions ratios.




Year-to-year variations of the merged sectoral CH$_4$ fluxes are presented in Fig. 9. Although systematic differences of absolute magnitudes exist between the SURF (or SURF+AIR) and GOSAT inversions, their changes are generally consistent
with each other. Interestingly, every sector contributed to the increase of CH$_4$ emissions for 2020–2022, but in different ways. One prominent feature is the increases in wetland and agriculture & waste emissions in 2020 (magnitudes of approximately 15 Tg CH$_4$ yr$^{-1}$ for both). Also, fossil fuel emissions are increased in 2020 by ca. 10 Tg CH$_4$ yr$^{-1}$, but the previous drop in 2019 is notable. Biomass burning emissions have two peaks in 2019 and 2021 (up to 10 Tg CH$_4$ yr$^{-1}$). In 2021, the SURF and SURF+AIR inversions showed a decrease of wetland emissions, but more biomass burning emissions in the northern high-
latitudes. In contrast, the GOSAT inversion estimated a more moderate decrease and increase for each of these, respectively. However, all the inversions agree with the increases in agriculture & waste emissions in the northern low-latitudes and in wetland emissions in the tropics, both of which contributed a large part of the global CH$_4$ emissions increase for 2020–2022.

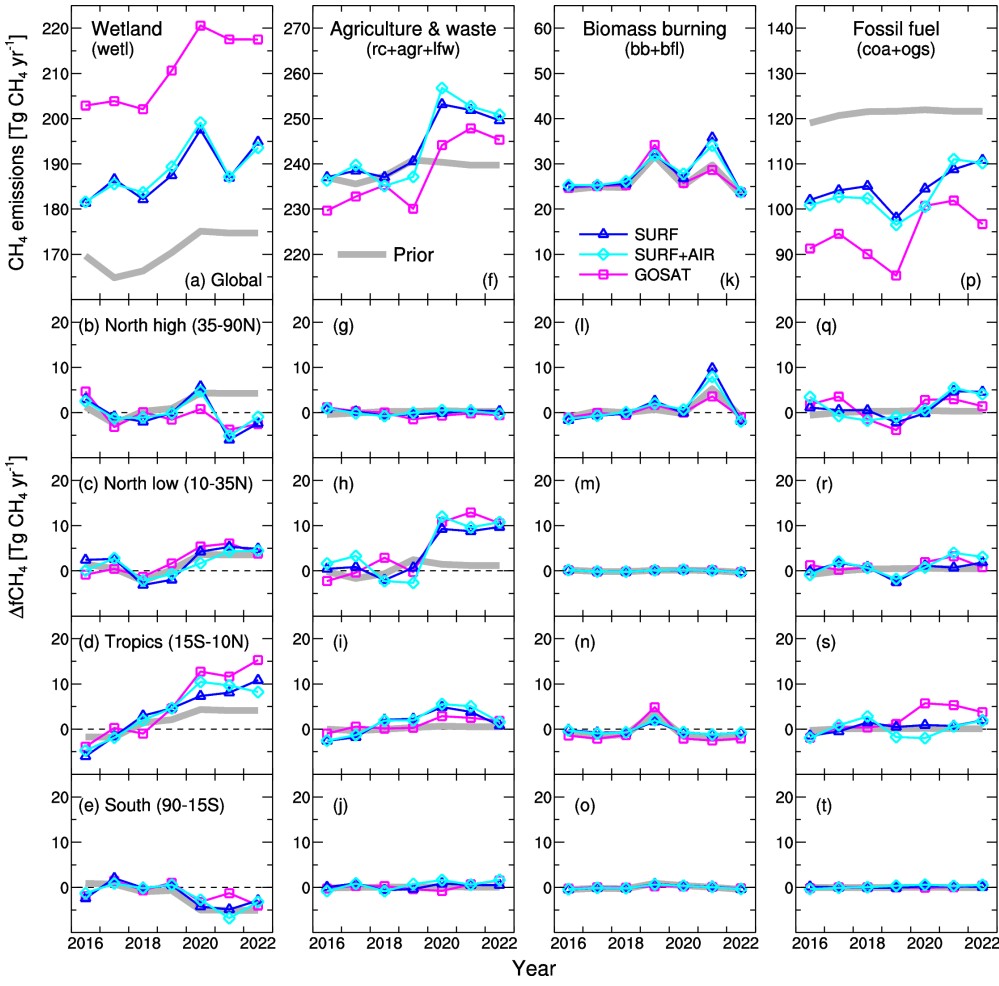



**Figure 9** Same as Fig. 5, but separated for sectoral emissions from wetland (wetl) (a–e), agriculture & waste (rc+agr+lfw) (f–j), biomass burning (bb+bfl) (k–o), and fossil fuels (coa+ogs) (p–t). Sectorial emissions are merged as shown in Table 1.

Figure 10 summarizes regional ΔfCH₄ increases for 2020–2022 by four sectors. The inversions consistently suggested that the emission increases in tropical South America and northern Southeast Asia were attributable to wetland and agriculture & waste (dominated by rice cultivation in the prior flux), respectively. The inversions also agree that the emission increase in South Asia was from both wetland and agriculture & waste sectors. However, the other emissions are estimated differently by the inversions. Biomass burning emissions are estimated to have increased in northern Eurasia and Boreal North America by the SURF and SURF+AIR inversions, but the increase is offset by decreases in Southeast Asia and other regions. For fossil fuel emissions, the GOSAT inversion suggested a large increase in contributions not only from the Asian regions but also from Central Africa.

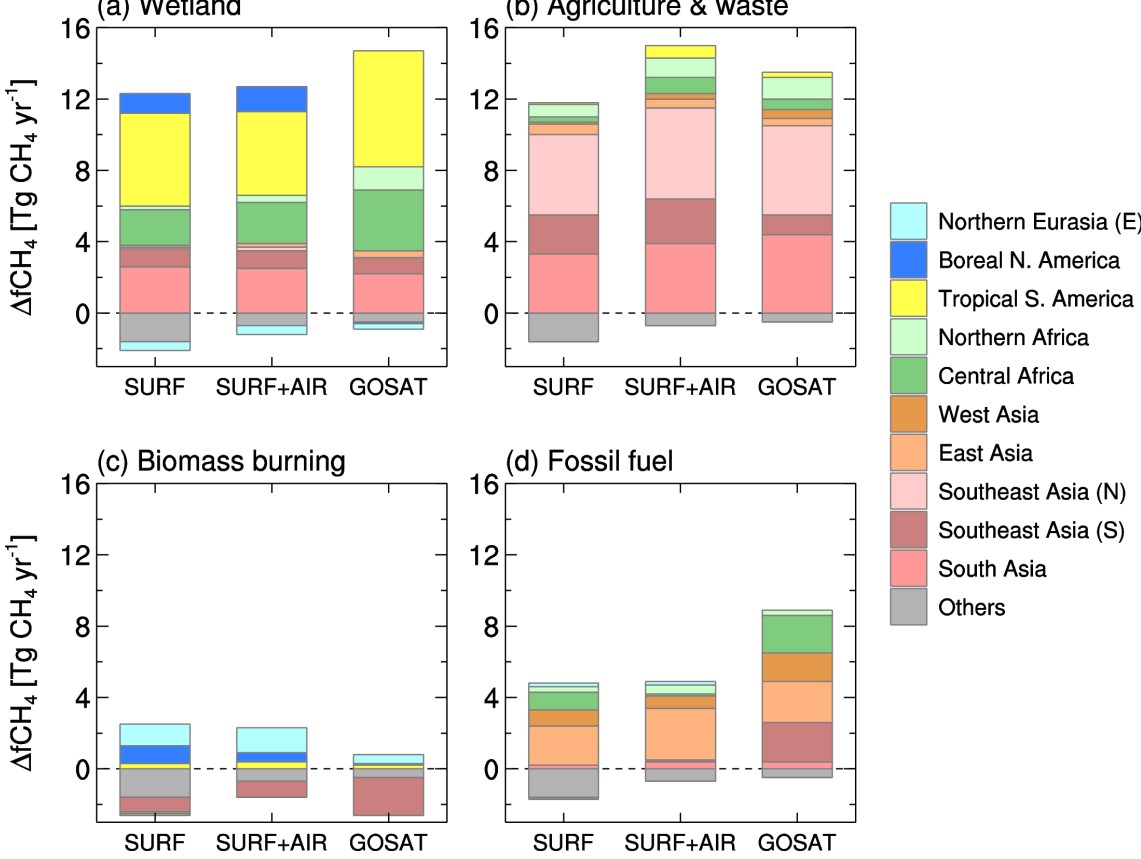

**Figure 10** Same as Fig. 7, but separated for sectoral emissions from wetland (wetl) (a), agriculture & waste (rc+agr+lfw) (b), biomass burning (bb+bfl) (c), and fossil fuels (coa+ogs) (d). Sectorial emissions are merged as shown in Table 1.




The aforementioned sectoral contributions have uncertainties, because in many regions, different sectoral emissions overlap or are close enough to well-mixed flux signals in the atmosphere. To assess uncertainties of sectoral contributions, we calculated posterior error correlations among the three major sectors: wetland, agriculture & waste, and fossil fuel emissions. Figure 11 shows the regionally integrated posterior error correlations between two of the three sectors for the three inversions.

South Asia, the biggest contributor to the emissions increase for 2020–2022 (Fig. 7), has the strongest anti-correlation between wetland and agriculture & waste emissions. This anti-correlation is particularly enhanced when aircraft or GOSAT data are used, probably because the aircraft observation or GOSAT observe well-mixed airmasses that cannot resolve wetland/agriculture emission signals. Furthermore, the other dominant contributors (northern Southeast Asia and tropical South America) also have notable anti-correlations between wetland and agriculture & waste emissions. Therefore, the

dominant increases of agricultural & waste emissions in northern Southeast Asia and wetland emissions in tropical South America (Fig. 10) might have some contributions by wetland emissions and agriculture & waste emissions, respectively. Nevertheless, because wetland or agriculture & waste emissions in these areas do not have notable anti-correlations with fossil fuel emissions, we can conclude that biogenic (wetland and/or agriculture & waste) emissions have dominantly contributed to the increase for 2020–2022.

The contribution of wetland emissions in Central Africa is also large (Fig. 10a), and its anti-correlations with the other emissions are small, indicating the robustness of the wetland contribution there. Fossil fuel emissions in East Asia, which have the largest contribution in this sector (Fig. 10d), have notable anti-correlations with wetland and agriculture & waste emissions (Figs. 11e and f), indicating the possibility of contributions from biogenic emissions. Error correlations of fire emissions, which are not shown in Fig. 11, are small compared to the abovementioned ones. A small negative error correlation

of −0.2 at most with wetland emissions was found in the east part of Northern Eurasia for the SURF+AIR inversion, while other areas/cases have negligible anti-correlations. This is probably because fires occur in a relatively small area, which makes it easy to separate them from other sector emissions (note that fluxes are optimized at each 1°×1°grid point, not in each aggregated region).



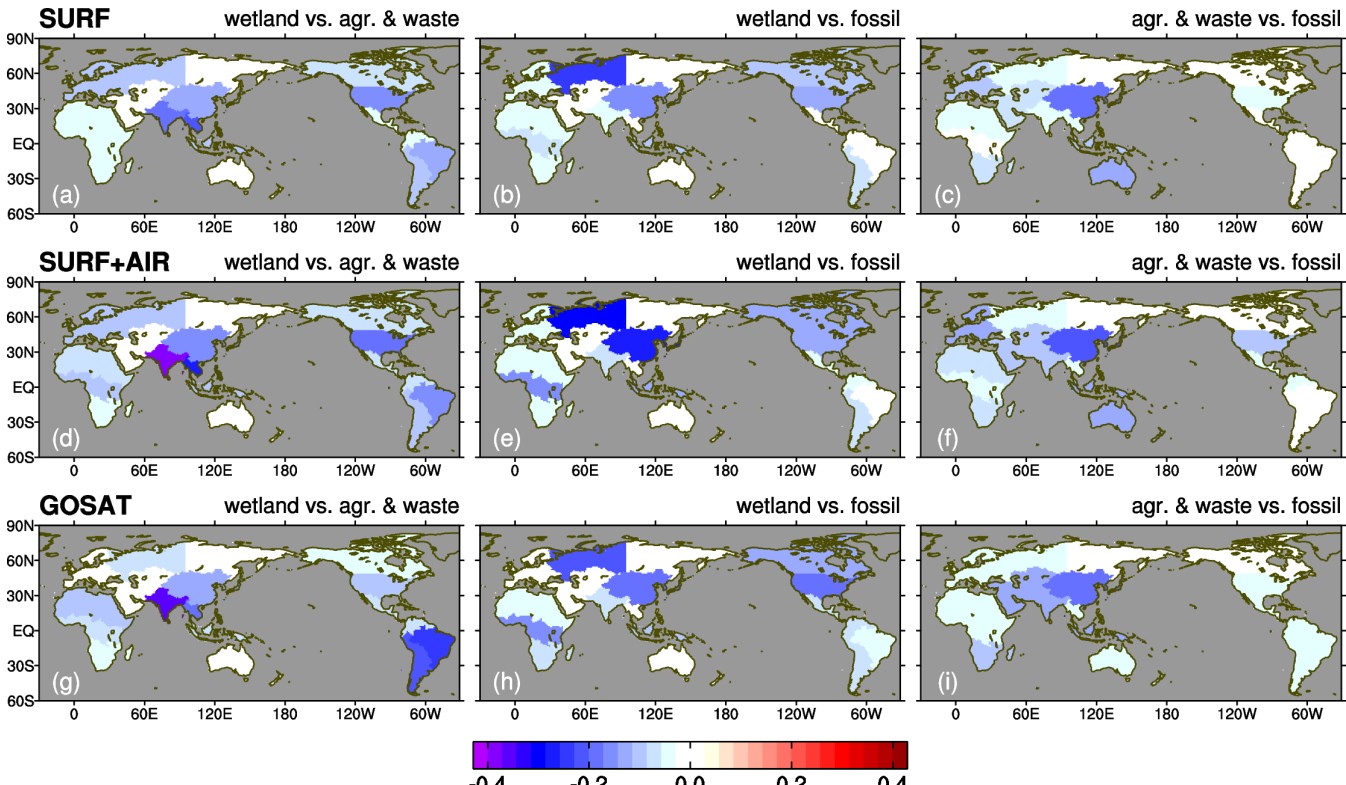

**Figure 11** Posterior error correlations between wetland and agriculture-waste emissions (left), wetland and fossil fuel emissions (center), and agriculture & waste and fossil fuel emissions (right) for the SURF (top), SURF+AIR (middle) and GOSAT (bottom) inversions. Error correlations are aggregated for each geographical region (Fig. 6).

## 4 Discussion

### 4.1 OH reduction due to the COVID-19 pandemic

This study investigated the surge of atmospheric $CH_4$ during 2020–2022 by the inverse analysis with NISMON-$CH_4$, which assumes that atmospheric OH abundance did not change during this period. However, we recognize that this is an optimistic assumption, especially for 2020. In fact, previous studies have suggested a significant contribution from the OH decrease as a result of the COVID-19 pandemic (Qu et al., 2022; Feng et al., 2023; Stevenson et al., 2022; Peng et al., 2022). Therefore, we performed a sensitivity test by reducing OH and accounting for the pandemic in 2020, although we used a simpler approach compared with other studies that considered atmospheric chemistry reactions with NOx. Details of the approach are described in Appendix C.

In the sensitivity tests of SURF and GOSAT, total $\Delta fCH_4$ in 2020 was reduced by 17 % and 29 % globally, respectively. Furthermore, those impacts appeared in the northern low-latitude and tropical regions, where notable increases





of emissions were found in the control inversions (Fig. C1). However, the reduction of OH we tested in this study is relatively large compared to other studies. Moreover, a recent study suggested much less OH reduction using multiple hydrofluorocarbon observations (Thompson et al., 2024). Therefore, our estimate of the effect of the OH reduction might be overestimated. Even with the reduced OH, northern Southeast Asia, one of the prominent contributors to the atmospheric $CH_4$ surge, still shows a notable emissions increase from 2019 to 2020. Furthermore, the reduced OH inversion with GOSAT still shows a notable emission increases in South Asia from 2019 to 2020. These results indicate that these Asian regions contributed to the surge of atmospheric $CH_4$ growth from 2019 to 2020.

### 4.2 Uncertainties in regional estimates

One notable feature of our inversion results is that the biogenic (wetland and agriculture & waste) emissions from Asia are the most important contributor to the increase of atmospheric $CH_4$ since 2020 (Fig. 7). However, similar inversion studies of Qu et al. (2022) and Feng et al. (2023) suggested a higher contribution from Africa. These previous inversions mainly used GOSAT data that were produced by the proxy method of the University of Leicester (GOSAT-UoL; Parker and Boesch, 2020), which has more data than the NIES GOSAT product we used (Fig. D1b). To examine influence of the different GOSAT products, we performed an additional inversion using the GOSAT-UoL data with the same inversion settings, but the period covered was only through 2021 because of data availability (Appendix D). In this inversion analysis, we obtained a notable increase of $\Delta fCH_4$ in Northern Africa for 2020–2021 as well as in Central Africa for 2020 (Fig. D2). Meanwhile, compared with the GOSAT-NIES inversion, $\Delta fCH_4$ is reduced in East Asia for 2020 and northern Southeast Asia for 2020– 2021, though the increase of $\Delta fCH_4$ in South Asia for 2020–2021 is retained or even enhanced for 2020. This result indicates that the increase of $CH_4$ emissions from Africa suggested by the previous studies is attributable to the use of GOSAT-UoL data, probably because GOSAT-UoL has flux signals from Africa that are not represented in GOSAT-NIES, surface, or aircraft data. In fact, as shown by the error reduction ratio in Fig. D2, GOSAT-UoL imposed strong constraints on flux estimates for the African regions. As shown in Fig. 7, the GOSAT-NIES inversion estimated larger emissions in Africa than the SURF and SURF+AIR inversions. Therefore, the larger emission increase from Africa is attributable to GOSAT itself, regardless of the product used. As of now, we cannot conclude which regional emission has made the largest contribution to the atmospheric $CH_4$ surge since 2020. The error reduction ratios by GOSAT-UoL are larger in the African and Asian regions than those of GOSAT-NIES (Fig. D2), but they are calculated under the assumption that observations are not biased. For evaluating these satellite products differences, we need to expand in-situ or flask observation networks, especially in the African regions; this would also be useful to investigate notable differences of atmospheric $CH_4$ between GOSAT and flask observations found in the tropics and southern latitudes (Fig. B1; similar differences are also found between GOSAT-UoL and flask observations (not shown)).

Meanwhile, Appendix D also highlights the importance of emissions from the Asian regions, using unique surface observations from NIES, which include ground-based flask samplings in the Asian countries (India, Bangladesh, and Malaysia) (Nomura et al., 2021) and ship measurements in the western Pacific and around Southeast Asia (Terao et al., 2011; Nara et al.,



2017). In fact, these observations provided greater confidence in flux estimates in the Asian regions by providing stronger
560 observational constraints (Fig. D2). Although omitting the NIES observations did not largely change the general features of
the ΔfCH₄ changes, Fig. D2 shows that the increase in 2021 was clearly attributed to the use of the NIES observations.
Furthermore, as shown by Fig. 6, aircraft data (which were uniquely used in the inversion in this study) supported the large
emissions increase from the Asian regions with additional strong constraints. The effectiveness of aircraft data in constraining
the estimates is attributable to active vertical transport, which is typical in these regions (i.e., the summer monsoon) (Niwa et
565 al., 2012, 2014, 2021).

### 4.3 Sectoral contributions

Whether from Africa or Asia, our inversions agree with previous inversions in that biogenic emissions dominated the
probable increase of CH₄ emissions. This large biogenic emission contribution is consistent with other studies that use the
stable CH₄ isotope ( $\delta^{13}$CH₄) measurements (Nisbet et al., 2023; Chandra et al., 2024). The expanded area of inundation,
570 which is probably related to the prolonged La Niña for 2020–2022, might have increased biogenic emissions in the northern
low-latitude areas (Feng et al., 2023; Lin et al. 2023). Detailed analyses on these sectoral contributions by comparing them
with meteorological parameters would provide insights into CH₄ emissions mechanisms. To this end, including the year of
2023 in the analysis period would be beneficial because the climate changed from La Niña to El Niño conditions in 2023. In
fact, the growth rate of atmospheric CH₄ seems to have decreased in 2023 (Lan et al., 2024). Furthermore, using observations
of the stable CH₄ isotope would also be beneficial (Lan et al., 2021; Chandra et al., 2024). Additional analyses focusing on
these climate condition changes are left for a future study.

The SURF and SURF+AIR inversions also suggested emission increases in the northern high-latitudes from wetlands
for 2020 (ca. 5 Tg CH₄ yr⁻¹) and from biomass burnings for 2021 (ca. 9 Tg CH₄ yr⁻¹) (Figs. 9 and 10). They probably can be
attributed to the Siberian heatwave in 2020 (Overland and Wang, 2021) and the boreal fires in 2021 (Zheng et al., 2023),
respectively. In fact, even though these emission increases are large enough to note, the decrease of wetland emissions in 2021
makes the contribution of the northern high-latitudes to the 2020–2022 surge minor (Fig. 7). Meanwhile, the GOSAT inversion
did not clearly reproduce emission increases in the northern high-latitudes, and this difference should be investigated in a
future study.

### 5 Conclusions

This study used the inversion method with multiple observational datasets to estimate probable emission increases that induced
the latest record-breaking surge of atmospheric CH₄ in 2020–2022. Using three different observational datasets (SURF,
SURF+AIR, and GOSAT), this study suggested that emissions in the tropics and the northern low-latitude areas notably
increased by 10–18 Tg CH₄ yr⁻¹ and 20 Tg CH₄ yr⁻¹, respectively, from 2016–2019 to 2020–2022. Specifically, the inversions
consistently estimated notable emission increases in tropical South America (5–7 Tg CH₄ yr⁻¹), central Africa (3–6 Tg CH₄



yr$^{-1}$), South Asia (6–7 Tg CH$_4$ yr$^{-1}$), and northern Southeast Asia (5 Tg CH$_4$ yr$^{-1}$). The emissions in tropical South America and central Africa showed gradual persistent increases for the analysis period (2016–2022) and they are mostly attributable to wetlands. The results also indicated that the two Asian regions (South Asia and northern Southeast Asia) contributed to the surge of atmospheric CH$_4$ with the sharp annual rises in their emissions from 2019 to 2020, and the elevated emissions continued until 2022. For these two regions, wetland and agriculture & waste sectors were estimated to be the largest

contributors to the increased emissions for the period, although notable anti-correlations of the posterior errors indicate that relative contributions from these two regions or these two sectors remain underdetermined. Because agriculture & waste emissions are derived from anthropogenic activities, the results of this study indicate a potential impact of direct emissions reduction measures on this sector for these two Asian regions.

The above inversion results are reliable through several reasons: (1) the spatiotemporal variations of posterior

atmospheric CH$_4$ mole fractions are improved from the prior ones in comparison with multiple observations and (2) the inversions with independent observations agree with each other for the above-mentioned emission increases (SURF(or SURF+AIR) and GOSAT). The flux estimates for the Asian regions are particularly noteworthy because the probable reduction of OH resulting from the pandemic-derived lockdown would not largely affect the flux estimates in Asia, as suggested by the sensitivity test results. Furthermore, the surface and aircraft observations, which were newly introduced in this study, provided

strong constraints and increased the confidence in the Asian flux estimates.

Other studies using the GOSAT proxy method data suggested the predominant role of emission increases in Africa. The results of this study cannot deny Africa as a possible source of the emissions increase, but they give prominence to the biogenic emissions in South Asia and northern Southeast Asia for the surge of atmospheric CH$_4$ from 2019 to 2020–2022. Because of the differences in flux estimates in the different satellite datasets, we need more elaborate networks of high-

precision in-situ and flask observations, especially in the tropical and low-latitude areas of Africa, South America, and Asia.

## Appendix A  Exterior penalty function method to avoid negative fluxes

Surface CH$_4$ fluxes from each sector are mostly one way; that is, fluxes other than soil uptakes are all positive (from the surface to the atmosphere), and soil uptake fluxes are all negative (from the atmosphere to the surface). In fact, this is also true for the prior flux data. However, it is not the case for posterior fluxes, because scaling factors or flux deviations optimized through

inversion may induce unrealistic negative fluxes (and positive ones for soil uptakes). To avoid such unrealistic fluxes, CH$_4$ inversions often use a numerical technique. For instance, Bergamaschi et al. (2009) transformed control variables with a "semiexponential" function.

In NISMON-CH$_4$, we use the exterior penalty function method (Sawada and Honda, 2021), which introduces an additional constraint with a so-called "penalty term" in the cost function. This penalty term $J_p$ is defined as


$$J_p(\boldsymbol{x}) = \lambda \sum_n \sum_i \left[ \max\{0, -f_{n,i}(\boldsymbol{x}_n)\} \right]^\alpha, \qquad (A1)$$



where $x$ is the control variable vector (i.e., scaling factors and flux deviations) and the indices $n$ and $i$ represent each flux sector and grid point, respectively. The flux operator $f_{n,i}$ calculates a flux value of the $n$th sector at the $i$th grid point from the control variables defined by each term of the right-hand-side of Eq. (1) (note that the minus before $f_{n,i}$ is omitted for soil uptakes to invert the sign). Combined with the conventionally defined cost function $J$ (similar to Eq. (1) of Niwa et al. (2022)), this

penalty term $J_\mathrm{p}$ leads to a constrained optimization problem (negative fluxes are avoided) as:

$$J_\mathrm{c} = J + J_\mathrm{p}, \qquad \text{(A2)}$$

which is used as the cost function in the 4D-Var iterative calculation instead of $J$. Because negative fluxes make the cost function extremely large, they are avoided in the optimal state, where the cost function reaches the minimum. In this study, the arbitrary parameters of $\lambda$ and $\alpha$ in Eq. A1 were set 1000 and 2, respectively, which were determined according to results of

practical optimization trials in terms of computational stability.

**Appendix B  Mean differences between observed and modelled atmospheric CH$_4$**

Here, we demonstrate how the modelled mole fractions of atmospheric CH$_4$ are consistent with observations before and after the inversions. Figure B1 shows mean differences between observed and modelled atmospheric CH$_4$ for surface, aircraft, and GOSAT observations for 2020–2022. These data are the same as those used in Fig. 2, but the offsets (the averages for 2016–

2019) are not subtracted; that is, a more direct comparison is conducted here. In general, the SURF, SURF+AIR, and GOSAT inversions are most consistent with the surface, aircraft, and GOSAT observations, respectively. This is not surprising, but it demonstrates that each inversion succeeded in optimizing atmospheric mole fractions as well as fluxes consistently with observations. However, when compared with independent observations, the inversions do not necessarily produce a better agreement with the observations than the prior fluxes do. This is attributable to errors in atmospheric transport, chemical loss

by OH, or in the measurements themselves. The persistent deviations of the GOSAT inversion from the surface observations are especially noticeable in the tropics and the southern latitudes (Fig. B1a), which can also be seen to a lesser extent in the comparison with the aircraft observations (Fig. B1b). Meanwhile, the SURF and SURF+AIR inversions largely deviate from the GOSAT observations in the tropics and southern latitudes (Fig. B1c). Given that in-situ and flask observations have much higher precision than satellite observations, this result indicates that the GOSAT observations have measurable biases in those

latitudes.





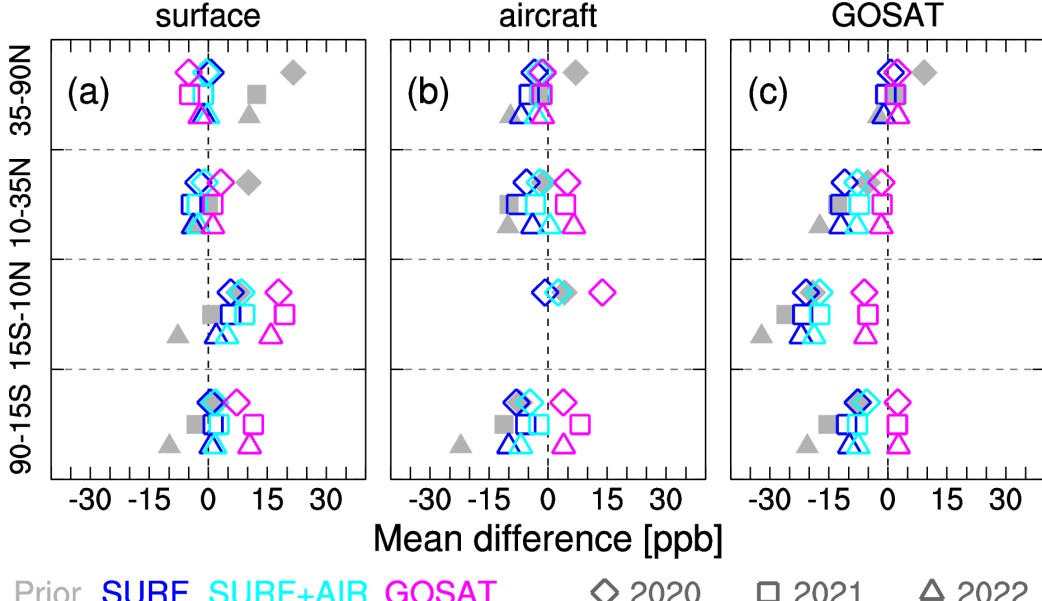

**Figure B1** Mean differences between observations and prior or posterior mole fractions of atmospheric $CH_4$ in northern high-latitudes (35–90°N), northern low-latitudes (10–35°N), tropics (15°S–10°N), and southern latitudes (90–15°S) for 2020 (diamond), 2021 (square), and 2022 (triangle). The mean differences are calculated for different observational types of surface (a) and aircraft (b) flask observations, and for the GOSAT data (c). The prior and posterior mole fractions are derived from atmospheric simulations of NICAM-TM with the prior (gray) and posterior fluxes: SURF (blue), SURF+AIR (cyan), and GOSAT (magenta), respectively.

## Appendix C  Inversions with reduced OH in 2020

To investigate the probable OH reduction resulting from the COVID-19 pandemic in 2020, we performed sensitivity tests for the SURF and GOSAT inversions. In these tests, we reduced the climatological OH data that were used in the control inversions according to the reduction of fossil fuel $CO_2$ emissions in 2020. Specifically, we used the fossil fuel $CO_2$ emissions from the gridded fossil emissions dataset (GridFED: Jones et al., 2021) and calculated their reduction ratios from 2019 to 2020 for each month and grid. Then, each calculated reduction ratio was applied to the OH field over the same grid below the 12th model layer (approximately 3 km above ground level), assuming that the reduction of NOx emissions and the consequent reduction of atmospheric OH occurred within the surface mixed layer at the same rate as that of $CO_2$ emissions. The global average of the resulting OH field is smaller by a maximum of 4 % in May than that of the climatological average, and the annually averaged reduction ratio is 2.5 %. This assumed OH reduction is larger than those of Peng et al. (2022) (1.6 %) and Qu et al. (2022) (1.2 %), but it is similar to that of Miyazaki et al. (2021) (4 % in May at a maximum). For the years other than 2020, we used the same climatological OH field.

Figure C1 shows the same regional $\Delta fCH_4$ changes as Fig. 5, but for the additional inversions with the reduced OH. In general, the boreal northern regions, which have less OH, are negligibly affected by the OH reduction for both inversions




(SURF and GOSAT). Meanwhile, estimated emissions were reduced by approximately 2 Tg CH$_4$ yr$^{-1}$ in temperate and tropical areas, except for tropical South America, which reduces the emissions increase in 2020. However, even with the OH reduction, northern Southeast Asia still shows a pronounced increase from 2019 to 2020. For South Asia emissions, SURF shows a

marginal increase, but GOSAT still shows a large increase from 2019 to 2020. In tropical South America, the reduced OH induced the largest emissions reduction (approximately 4 Tg CH$_4$ yr$^{-1}$), producing a notable drop from the before and after years. Globally, total ΔfCH$_4$ in 2020 decreased by 17 % and 29 % for the SURF and GOSAT inversions, respectively.

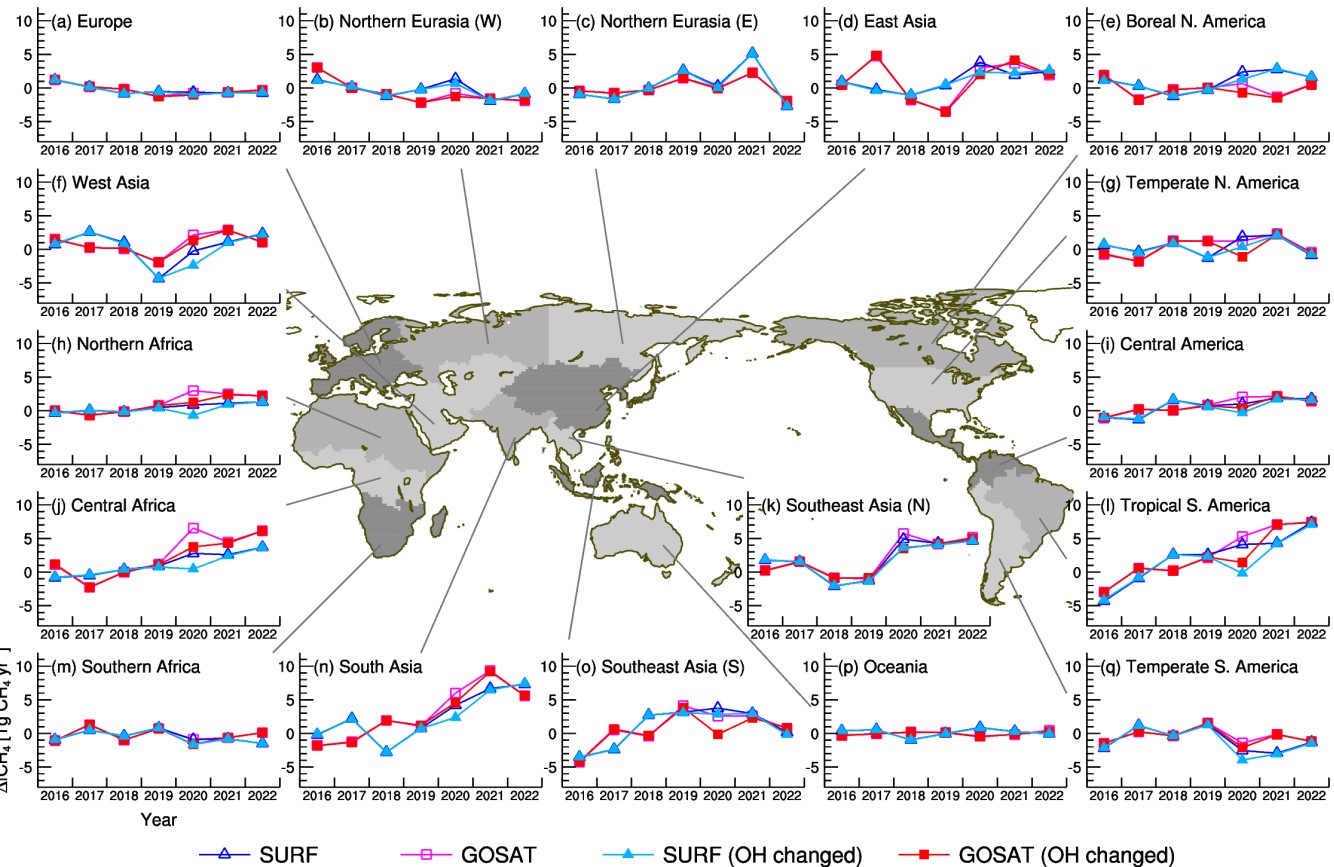

**Figure C1** Same as Fig. 6, but including the sensitivity tests with OH reduced in 2020 for the surface (light blue closed triangles) and GOSAT (red closed squares) inversions.

## Appendix D  Inversions without the NIES observations and with the University of Leicester GOSAT proxy data

We performed two additional inversions using different observational networks. One uses the same surface observations but

excludes the NIES observations (SURF w/o NIES; Fig. D1a). The NIES observation network includes flask samplings in



South and Southeast Asia, ship measurements in the Asia-Pacific regions, and in-situ measurements using towers in Siberia, but those NIES data (except for Siberian) are not included in NOAA GLOBALVIEWplus, which is a major dataset used in other inversion studies, which is a major difference between our study and the others. The second inversion employed the University of Leicester (UoL) version 9.0 GOSAT proxy data (Parker and Boesch, 2020) (GOSAT-UoL) (Fig. D1b).

In this study, we used the NIES GOSAT product, which is produced by the full-physics retrieval method (Yoshida et al., 2011, 2013). Meanwhile, the proxy data are produced by a method that uses modelled $CO_2$ mole fractions as a proxy to retrieve $XCH_4$, which is less affected by aerosols and clouds. As shown by Fig. D1b, there are more data available with the proxy method than with the full-physics method (Fig. 1c), particularly for Africa and South America. At the time of this study, the GOSAT-UoL data were available through the end of 2021. Therefore, the inversion with GOSAT-UoL was performed for

the period until 2021.

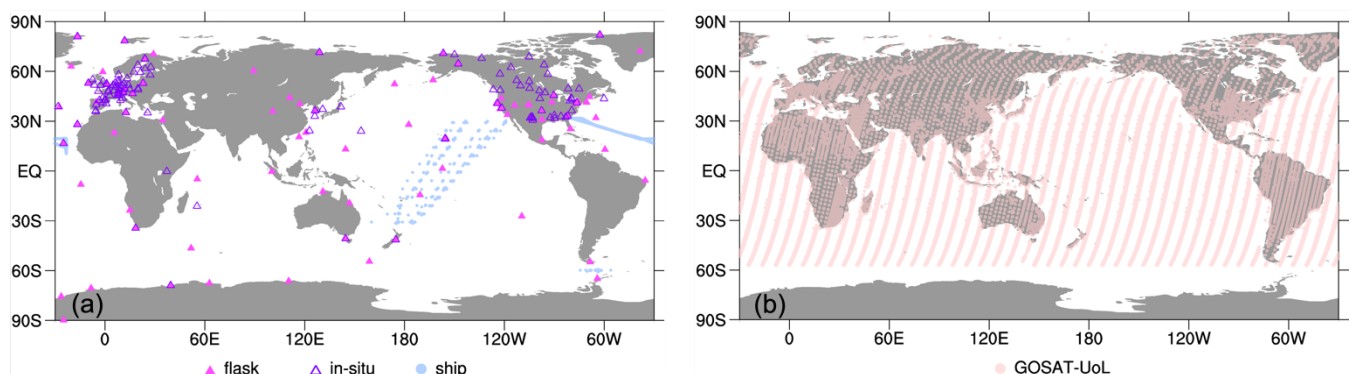

**Figure D1** Same as Figs. 1a, but the NIES observations are excluded (a). The GOSAT proxy data from the University of Leicester obtained during 2020 (b).


Figure D2 shows the same temporal pattens of $\Delta fCH_4$ for each region as Fig. 5, but the additional inversion results are presented. In general, the additional inversions of SURF w/o NIES and GOSAT-UoL show temporal variations similar to those of the corresponding control inversions, although there are notable differences in some regions. The SURF w/o NIES inversion estimated smaller $\Delta fCH_4$ increases in northern Southeast Asia and South Asia for 2021, but it still showed elevated

$\Delta fCH_4$ in 2020 and 2022. Compared to the GOSAT inversion, the GOSAT-UoL inversion shows remarkably large $\Delta fCH_4$ increases in Northern Africa for 2020 and 2021, with smaller $\Delta fCH_4$ in East Asia and northern Southeast Asia. The observational constraint (represented by the error reduction ratio) is also different from the original inversion. The SURF w/o NIES inversion shows weaker observational constraints than those of SURF in the Asia and Oceania regions, indicating that the NIES observations have strong constraints in flux estimates for these regions. Meanwhile, the GOSAT-UoL inversion

shows stronger observational constraints than the GOSAT inversion everywhere, and this is more pronounced in the tropical regions (almost doubled). These stronger constraints are attributed to the larger amount of data in GOSAT-UoL (Fig. D1b).



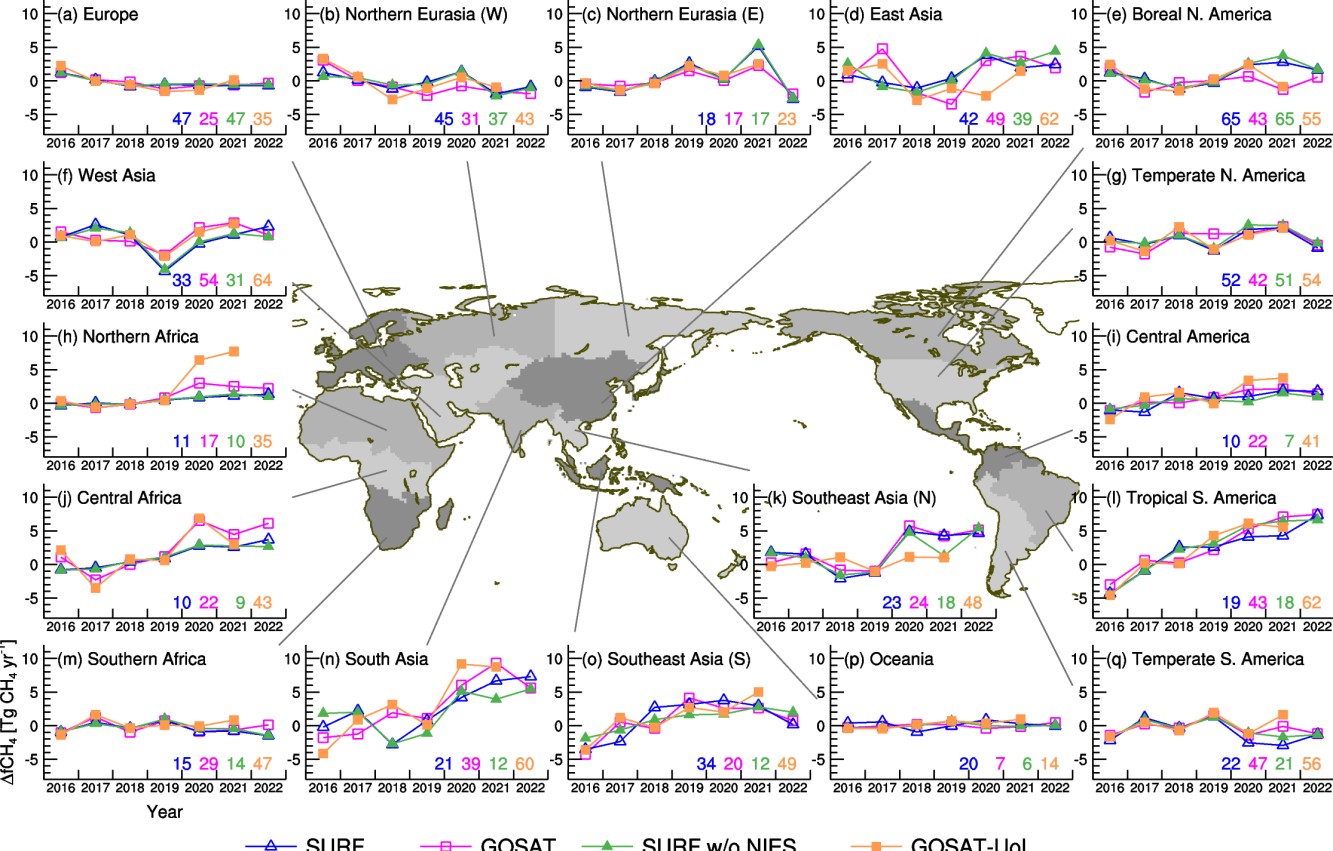

**Figure D2** Same as Fig. 6, but the additional inversions using surface observations excluding the NIES observations (green closed triangles)
and using the University of Leicester (UoL) GOSAT proxy data (orange closed squares) are shown. Numbers in each panel denote error
reduction ratios.

**Data availability**

In-situ and flask observations of atmospheric CH4 can be obtained from NOAA ObsPack GLOBALVIEWplus (Schuldt et al.,
2023a) and NOAA ObsPack NRT (Schuldt et al., 2023b), which includes ICOS (European CH4 ObsPack: ICOS RI et al.,
2023). The in-situ and flask observations of NIES and collaborative networks (the Asian sites, VOS, CONTRAIL, JR-
STATION and the Siberia aircraft) are available from the NIES Global Environmental Database (GED:
https://db.cger.nies.go.jp/ged/en/index.html) (JR-STATION and CONTRAIL data are also included in ObsPack
GLOBALVIEWplus). The aircraft data of Tohoku University are available from the World Data Center for Greenhouse Gases
(WDCGG: https://gaw.kishou.go.jp/). NIES GOSAT data are available from the NIES GOSAT Data Archive Service (GDAS:





https://data2.gosat.nies.go.jp/index_en.html) and the University of Leicester GOSAT data are available from the CEDA Archive (https://dx.doi.org/10.5285/18ef8247f52a4cb6a14013f8235cc1eb).

**Author contributions**

YN designed and conducted the inversion analyses. YT, YT, TU, MS, TM, SN, HN, HT, HM, YY, SM, KS, KT, YS, HM, DG, XL MH, TB, LC, JN, and IXR provided in-situ and flask observations of atmospheric $CH_4$. TS and YY provided GOSAT-NIES observations. TS also developed the observation operator of the satellite data in NISMON-$CH_4$. AI provided the VISIT $CH_4$ flux data. KI, KI and RF reviewed and commented on the paper. YN prepared the manuscript with contributions from all co-authors.

**Competing interests**

The authors declare that they have no conflict of interest.

**Financial support**

**Financial support and Acknowledgements**

This study has been supported by the Environment Research and Technology Development Fund of the Environmental Restoration and Conservation Agency (JPMEERF21S20800 and JPMEERF24S12200), provided by Ministry of the Environment of Japan, and partially supported by the NIES Research Funding (Type A), the NIES Climate Change and Air Quality Research Program, and the JSPS KAKENHI (grant no. JP22H05006, JP23H00513). The inversion system used in this study has been developed under the cooperative research between MRI and NIES, and its numerical calculations were
performed on supercomputer systems at MRI and NIES (FUJITSU PRIMERGY CX2550M5 and NEC SX-Aurora TSUBASA, respectively). YN is grateful to all the staff working on those supercomputer systems. YN also thanks Prof. Masaki Satoh and others at the University of Tokyo, JAMSTEC, RIKEN, and NIES for developing NICAM. The observational projects of CONTRAIL and NIES VOS and JR-STATION are financially supported by the research fund of the Global Environmental Research Coordination System of the Ministry of the Environment, Japan (E0752, E1253, E1254, E1652, E1752, E1851, 745 E2151, E2251, E2351, E2452). The flask air sampling observations at Asian stations were supported by the Environment Research and Technology Development Fund of the Environmental Restoration and Conservation Agency



(JPMEERF20152002 and JPMEERF20182002). The observations from the CONTRAIL project are conducted with support from Japan Airlines, JAMCO, and the JAL Foundation. The NIES VOS programme has been conducted in collaboration with Toyofuji Shipping Co., Ltd and Kagoshima Senpaku Co., Ltd. The CH$_4$ observations of Tohoku University over Japan are conducted in cooperation with J-Air Co. Ltd. and Japan Airlines. GOSAT mission is promoted by the Ministry of the Environment Government of Japan, NIES, and JAXA (Japan Aerospace Exploration Agency). Our appreciation is also extended to the many research groups contributing to the NOAA ObsPack datasets. Individual acknowledgements for ObsPack data are as follows. Atmospheric CH$_4$ observations at Lägern-Hochwacht are provided by Empa (Swiss Federal Laboratories for Materials Science and Technology) and those at Mount Kenya are a joint effort of the Kenya Meteorological Department, Empa and MeteoSwiss and the WMO Global Atmosphere Watch Programme. Atmospheric CH$_4$ observations at Jungfraujoch are supported by the Swiss Federal Office for the Environment and ICOS Switzerland (ICOS-CH) (Swiss National Science Foundation, grants 20FI21_148992, 20FI20_173691, 20F120_198227). The ICOS station Observatoire de l'Atmosphère du Maïdo (RUN) is a Belgian – France collaboration project and operated through a collaboration between the Royal Belgian Institute for Space Aeronomy (BIRA-IASB) and the following French partners: Commissiarat à l'Energie Atomique et aux Energies Alternatives (CEA), Centre National de la Recherche Scientifique (CNRS), Université de Versailes Saint Quentin-en-Yvelines (UVSQ), Université de La Réunion (UR). In Belgium, it has been financially supported since 2014 by the EU project ICOS-Inwire (Grant agreement ID 313169), the ministerial decree for ICOS (FR/35/IC1 to FR/35/C6) and ESFRI-FED ICOS-BE project (EF/211/ICOS-BE). Atmospheric CH$_4$ observations from Sweden have been provided by ICOS Sweden, funded by the Swedish Research Council as a national research infrastructure. The authors are grateful to M. De Mazière, M. K. Sha, N. Kumps and C. Hermans (BIRA-IASB) and J.-M. Metzger (UR) for their contributions to the labelling process, daily operations, and management of the station.

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
