# Peer review of "Multi-observational estimation of regional and sectoral emission contributions to the persistent high growth rate of atmospheric CH4 for 2020–2022"

_EGUsphere, 2024_

## Author Response (AR1)

Reviewer #1

This manuscript presents an inverse modelling system for estimating global methane emissions. It is used to study the causes of the accelerated methane increase during 2020-2022. Compared with previous studies, a larger set of surface and aircraft measurements is used, extending the data coverage in south and east Asia, in addition to the use of alternative GOSAT retrieval datasets. The results highlight the role of increasing Asian emissions in the global growth rate enhancement during this period, attributed mostly to increases in agricultural emissions. The sensitivity of inversion-estimated emissions to the observational datasets is an important – although not unexpected – finding. This study points to a trade-off between African and Asian emission increases depending on the data that are used, which makes an important contribution the scientific understanding of the causes of the global emission increase.

The manuscript is well written. Provided that the few points raised below are sufficiently well addressed I see no reason to uphold publication.

We are grateful for your time to review our paper and for giving us fruitful comments and suggestions. Our replices to the comments and modifications are described below with current line numbers.

GENERAL COMMENTS

The method section describes how posterior uncertainties are quantified. However, besides posterior flux covariances and uncertainty reduction very little use is made of posterior uncertainties. How do posterior uncertainties compare with the differences that are found between the different inversions? How about the significance of the most important flux deviations from the prior that are used to explain the 2020-2022 growth rate anomaly? Some of the plots miss error bars.

We agree with your comments on the posterior errors. In order to show absolute values of the posterior errors, we added the global totals of the errors as well as flux totals in Table 1. Furthermore, we also inserted bar plots presenting regional errors in Fig. 6. They could show how much uncertain each regional or sectoral emissions are compared to others. However, those abosolute values are small compared to the differences among the three inversions. In addition, they are also smaller than an inversion ensemble spread (e.g., Saunois et al. 2024). This indicates that these posterior errros cannot be considered as practical uncertainties of the inversion. Therefore, we did not put those posterior errors in the time series plots as error bars.

According to these modifications in Table and Figures, we added texts as below.

"The annual global totals and their integrated errors of the prior fluxes are presented in Table 1. "

[Line 178]

"Despite such differences among the posterior fluxes, the three inversions showed the same tendency of sectoral emission changes with respect to the prior data, such as larger wetland and rice cultivation emissions, and smaller coal mining and oil/gas emissions (Table 1). The errors of those emissions were reduced with respect to the prior ones, indicating that those emission changes were constrained by observations. However, it should be noted that the posterior errors are generally smaller than the differences among the three inversions. In addition, they are also smaller than an inversion ensemble spread (e.g., Saunois et al. 2024). Therefore, those calculated posterior errors cannot be considered as practical uncertainties of the inversion." [Lines 362–368]

Increases in emissions over Africa and southeast Asia are discussed, which have been attributed increases in natural wetlands and agriculture. However, it is not clear to which extend these increases are in the a priori fluxes already. A priori emission estimates in zonal bands are presented that give some indication, but it is unclear whether those differences are representative for what is found for the regions that are used in the sectorial bar graphs.

We added the prior bars in Figs. 7 and 10. Although the prior flux data already have emission increases, they are smaller than those of the posteior ones and limited only for wetlands.

The following text was added in the main text:

"Only for wetlands, notable increases are estimated by the prior data, in which the VISIT data for 2020 were repeatedly used for 2021–2022." [Lines 515–516]

The sensitivity to observational datasets and their spatial coverage raises the question whether the size of regional observational constraints could drive the differences in the outcomes of different inversions. If Asian data are added, the importance of Asian emissions increases, if data over Tropical Africa are added (i.e. proxy-method GOSAT retrievals) the importance of African emissions increases. It could be coincidence but might also be a symptom of sampling bias. It would be useful to add a data thinning experiment to distinguish between the extra information on methane emissions that new measurements bring versus the impact of their added observational constraint.

We understand the raised question; it should be clarified before concluding which emissions should have impacted the global increase of atmospheric $CH_4$. However, we also think that a thining data experiment only cannot elucidate that, because the extra information the new measurements brought versus the impact of their added observational constraint are not independent of each other. Furthermore, balancing with observations in other areas would also affect the results. In addition, we should also consider the effect of the observation-model mismatch error covariance too. If we added new observations, but assigned large errors, they would not impact flux estimates so much.

Nevertheless, we designed the experiment so that observational constraints should not get too strong, which was achived by the observational weighting $r_i$ introduced to the observation-model mismatch error covariance (Eq. 2). This works as data thining.

**SPECIFIC COMMENTS**

line 30: "increase" compared to what? It misses a notion of the extent to which this is expected or not given the a priori fluxes.

Thank you for pointing out that. We modified it as "increases from 2016–2019 to 2020–2022". [Line 31]

line 35: Agreement was found between what?

We modified it as "Agreement was found in the sectoral estimates of the three inversions" [Line 36]

line 119: How do you mean 'derived'? From what?

We elaborated it by replacing "deribed but modified from" with "reduced by 8% with respect to". [Line 126]

Equation 1: Parentheses are missing indicating the limits of the sum over i processes (that is only for a part of the equation, but it is unclear which part). Why are some processes corrected using delta-alpha and others using delta-f? This treatment makes an important but unexplained difference. Does delta-f cover grid boxes for which the corresponding f has zero emissions?

We added square brackets to make the range of the summation clearer. The reason why we mix delta-f and delta-alpha for the optimizing parameters is described in the next paragraph of Eq. (1) [lines 166–170]. To make it more noticeable, we add (delta-alpha) and (delta-f) after "the scaling factors" and "the flux deviations", respectively. As pointed out, delta-f could change fluxes even where f has zero emissions.

Sect. 2.2: What spatial and temporal error covariances are assumed of the 1x1 degree a priori monthly and annual fluxes?

For the delta-alpha parameters, we assumed no spatiotemporal error correlation. Meanwhile, for the delta-f parameters, we made the error covariance matrix from an ensemble, which are derived from a long-term prior simulation data. In this method, not only variance but also covariances were calculated from the ensemble. However, they are localized in space by a Gaussian function to damp erroneous correlations in remote areas. Here, we also assumed no temporal correlation.

To elaborate how to construct the prior error covariance matrix, we modified the last paragraph of Section 2.2. [Lines 171–178]

line 163: How large are the wetlands, rice, soil uncertainties derived from VISIT?

We added those prior uncertainties derived from VISIT in Table 1.

line 172: How about the temporal coverage of data that have been used? Is the sampling network changing over time?

In fact, some in-situ/flask observations stopped during the analysis period. Furthermore, aircraft data are more sporadic. Such inhomogeneous data could affect flux estimates in inversion. Meanwhile, GOSAT observations are more constantly obtained, though the data availability differs by seasons. Therefore, comparing inversions that used those different types of observations independently like this study would make up for each other's deficiencies.

We added sentences as follows:

", which could affect flux estimates in inversion."   [Lines 208–209]

"Although the data availability differs by seasons, the GOSAT observations have been more constantly obtained from one year to another than the in-situ or flask observations." [Lines 250–252]

line 187: 'deemed to be comparable' within what accuracy?

We elaborated it as

"(the difference is about 0.5 ppb (Fujita et al., 2018))" [Line 202]

Equation 2: How does balancing of data constraints work out for the observational weights of surface, aircraft and GOSAT data? How do the corresponding terms in the cost function compare?

The balancing parameter of $\beta$ was determined so that $X^2$ should be less than 1.

We modified the text about $\beta$ accordingly. [Lines 236–237]

Line 307: How does the inversions performance evaluation in Figure 2 distinguish between data that are or are not used in the inversion?

Because the GOSAT inversions did not use any in-situ/flask observations, for the GOSAT inversion, the comparisons with surface and aircraft data can be considered as evaluation with independent observations. Conversely, the GOSAT data are independent for the SURF and SURF+AIR inversions.

Figure 6: uncertainty reductions are a % reduction?

Thank you for pointing out it. We added % in the caption of Figure 6.

Appendix A: Using the method that is presented reduces the likelihood of negative emissions but does not prevent that negative emissions might happen. To which extent is this still the case?

As you pointed out, this scheme cannot avoid negative values perfectly. However, unavoidably generated negative values were at most three orders of magnitude smaller than positive values in the experiments. We added this explanation at the end of Appendix A. [Lines 688–690]

Appendix B: line 644: It is mentioned that GOSAT retrievals are biased, but this need not be the case. There could also be an inconsistency between modelled surface and total column mixing ratios due to a transport model problem. I doubt that comparisons between GOSAT and TCCON show this bias. Past studies that used GOSAT struggled with this too, but concluded that the problem was probably more a model problem than a retrieval problem.

We consider that the biaseas are not caused by the model, because we see similar biases of the GOSAT inversion both at the surface and in the free troposphere (Figs. B1a and B1b). Those consistent biases suggest that a problem due to vertical transport or chemical loss in the model is less likely.

To elaborate the discussion, we modified the last part of Appendix B as

"Those differences consistently existing at the surface and in the free troposphere may not be contributed by vertical transport or chemical loss in the model. Given that in-situ and flask observations have much higher precision than satellite observations, this result indicates that the GOSAT observations have measurable biases in those latitudes. Meanwhile, the SURF and SURF+AIR inversions largely deviate from the GOSAT observations in the tropics and southern latitudes (Fig. B1c), which is consistent with the GOSAT inversion results." [Lines 703–707]

Nevertheless, further investigation is needed to clarify the cause of those differences, which is left for a future study.

Appendix C: The description of the method is clear, but it would be helpful add a map of what the resulting OH reduction looks like?

As expected from the method, the map is showing that OH reductions are concentrated in the mid-latitude, where fossil fuel emissions are largely existing. Because that OH reduction map was made based on a very simple assumption (completely correlated with $CO_2$ emissions reduction, only within boundary layer) and must be different from that of more probable OH reduction, we did not add the map to avoid giving a wrong impression.

TECHNICAL CORRECTIONS

line 599: "for several reasons" instead of "through several reasons"

Thank you for your correction. We modified it. [Line 655]

**Citation**: https://doi.org/10.5194/egusphere-2024-2457-RC1

Reviewer #2

The manuscript entitled "Multi-observational estimation of regional and sectoral emission contributions to the persistent high growth rate of atmospheric CH4 for 2020–2022" by Yosuke Niwa and coworkers presents a detailed description of global CH4 fluxes estimated from atmospheric observations and an inverse modelling framework. Special attention is given to the recent rise in atmospheric methane growth rates. By employing an emission optimisation (inversion) by region and source sector Niwa et al. present the most likely drivers of the recent increase in CH4 emissions. The applied inversion tool and the analysis are sound and state of the art, the study appropriately addresses the limitations and uncertainties of the approach. Presentation of results is clear and concise. At this point I only have minor suggestions for modifications and additional clarifications.

We are grateful for your time to review our paper and for giving us fruitful comments and suggestions. Our replices to the comments and modifications are described below with current line numbers.

General comment

Although the study carefully scrutinises the main results by presenting several sensitivity inversions (different observational constraint, OH impact) and analysing posterior covariance between regions and sectors, more attention should be given to the posterior uncertainties themselves. None of the plots contains uncertainty estimates on any of the emission time series, nor are any uncertainty statements given in the text when emissions for a given region, sector or global total are discussed. The estimated posterior uncertainties could easily be employed to analyse the statistical significance of the observed, step-wise, increase in CH4 emissions after 2020 and support statements made about the equivalence of results obtained from different inversions. A discussion of uncertainty reduction was used to showcase which regions/sectors were well constraint by the observations, but the additional use of the absolute posterior uncertainties could largely enhance the discussion.

We agree with your comments on the posterior errors. In order to show absolute values of the posterior errors, we added the global totals of the errors as well as flux totals in Table 1. Furthermore, we also inserted bar plots presenting regional errors in Fig. 6. They could show how much uncertain each regional or sectoral emissions are compared to others. However, those absolute values are small compared to the differences among the three inversions. In addition, they are also smaller than an inversion ensemble spread (e.g., Saunois et al. 2024). This indicates that these posterior errros cannot be considered as practical uncertainties of the inversion. Therefore, we did not put those posterior errors in the time series plots as error bars.

According to these modifications in Table 1 and Fig. 6, we added texts as below.

"The annual global totals and their integrated errors of the prior fluxes are presented in Table 1. "

[Line 178]

"Despite such differences among the posterior fluxes, the three inversions showed the same tendency of sectoral emission changes with respect to the prior data, such as larger wetland and rice cultivation emissions, and smaller coal mining and oil/gas emissions (Table 1). The errors of those emissions were reduced with respect to the prior ones, indicating that those emission changes were constrained by observations. However, it should be noted that the posterior errors are generally smaller than the differences among the three inversions. In addition, they are also smaller than an inversion ensemble spread (e.g., Saunois et al. 2024). Therefore, those calculated posterior errors cannot be considered as practical uncertainties of the inversion." [Lines 362–368]

Specific comments

Abstract: Consider an alternative start that gives a bit more room for setting the stage. Something like: "Atmospheric methane (CH4) growth rates reached unprecedented values in the years 2020-2022. In order to identify the main drivers of this increase, we present results from an inverse modelling study estimating regional and sectoral emission contributions for the period 2016 to 2022. Three inverse estimates based on different sets of atmospheric CH4 observations (surface observations only, surface and aircraft observations, GOSAT satellite observations) consistently suggest notable emission increases from 2016-2019 to 2020-2022: ... "

We appreciate your suggestion. We have incorporated the suggested sentences in the beginning of Abstract. [Lines 28–31]

Abstract, following line 37: I am missing a discussion of the fossil emission trends here. Fig 10 shows a considerable (though smaller increase) as well for different Asian regions. I think this is worth mentioning in the abstract as well.

The increase of the fossil fuel emissons is small compared to that of the total biogenic emissions. As dicussion was made almost on the biogenic emissions in the main text, we keep to focus on biogenic emissions in Abstract too.

Nevertheless, as suggested, it is worth discussing fossil fuel emissions, we added discussion in the main text as

"For fossil fuel emissions, the GOSAT inversion suggested a large increase in contributions not only from the Asian regions but also from Central Africa (more than 8 Tg $CH_4$ $yr^{-1}$ in total). Meanwhile, the SURF and SURF+AIR inversions showed moderate increases of about 4 Tg $CH_4$ $yr^{-1}$, which are mostly from East Asia." [Lines 523–526]

and

"Following those biogenic emissions, the inversions suggested increases of fossil fuel emissions especially from Asian regions and Central Africa (Fig. 10). However, they were largely contributed by the recovery from the drop in 2019 (Fig. 9), whose cause is unclear at this moment. Furthermore, the increase of the fossil fuel emissions for 2020–2022 could be partly contributed by misallocation of biogenic emissions, because East Asia, which is the most contributor of this sector (Fig. 10), has anti-correlations between fossil fuel emissions and wetland and agriculture & waste emissions (Fig. 11)." [Lines 626–630]

Meanwhile, we modified Abstract to meet the limitation of the 250-words as follows:

"Agreement was found in the sectoral estimates of the three inversions in the tropics and northern low-latitudes, suggesting the largest contribution of biogenic emissions. " [Lines 36–37]

We also deleted specific numbers of regional emission increases and the description that the constraints of the surface and aircraft observations were comparable to or 1.5 times stronger than GOSAT constraints.

L37f: What is this quantification of constraint based on? The uncertainty reductions? Instead of only discussing the difference in constraint, I suggest to mention the general differences in GOSAT vs SURF inversions in terms of spatial and sectorial allocation.

That is based on the uncertainty reduction. We modified and also simplified this sentence because of the word limitation as

"Uncertainty reductions demonstrate that the flux estimates in Asia are well constrained by surface and aircraft observations." [Lines 37–38]

In fact, the differences between GOSAT and SURF/SURF+AIR inversions are interesting, but the main target of this study is to investigate the recent growth of atmospheric $CH_4$. In this view point, the agreement of emission increase estimates in those different inversions is the most imporant fact. Therefore, we kept not mentioning the difference between the GOSAT and SURF/SURF+AIR inversions in Abstract.

L39f: The statement on OH impact is not well formulated. Nor is the analysis in section 4.1 very detailed. I suggest updating after a revision of section 4.1 (see comment below).

We elaborated the text using numbers as follows:

"Furthermore, a sensitivity test with the probable reduction of OH radicals showed smaller emissions by up to 2–3 Tg $CH_4$ $yr^{-1}$ in each Asian region for 2020, still suggesting notable emission contributions." [Lines 38–40]

L54f: Sentence somewhat convoluted. Consider rephrasing. My suggestion: "In particular, CH4 has recently attracted global attention because due to its short lifetime, the mitigation effect on global warming when reducing its emissions occurs sooner than when reducing CO2 emissions. Hence, ambitious reduction targets were envisaged in the Global Methane Pledge for the coming years." Furthermore, a reference for the Global Methane Pledge and a more quantitative statement of its targets would underline the statement.

We appreciate your detailed suggestions. We took all of the suggestions. [Lines 54–58]

L65f: Li et al. (2023) report reductions for Jan – Apr for 2022. Main northern hemispheric sink will be in summer. Considering short NOx lifetimes I wonder how much impact can then be expected if emissions returned to previous levels for the rest of the year. Was this considered in the OH sensitivity run?

We agree that that NOx emissions reduction in China might not have affected $CH_4$ sink largely. In our study, we performed the OH sensitivity test only for 2020, when the NOx emission reduction effect seems to be the largest in 2020–2022. To add the information of the $CO_2$ redcution in China (January – April), we modified the following sentence (see the reply to the next).

L67: Sentence unclear: emissions of what? Contribution to what? Global NOx emissions to global OH levels?

We modified the sentence to clarify them as:

"However, that OH reduction effect was limited in space (not global) (Liu et al., 2023) and time (only for January–April) (Li et al., 2023), suggesting a continued contribution of $CH_4$ emissions." [Lines 68–69]

L90f: What does 'multidirectional analysis' refer to here? The sensitivity inversions carried out in the present manuscript or something else?

To clarify this, we modified the sentence as

"This kind of multi-observation analysis as carried out by Lin et al. (2024) is imperative to infer the observational uncertainties." [Lines 92–93]

L97ff, last paragraph of Introduction: Please include cross-references to the following sections where details on the mentioned models/analysis can be found.

We added the corss-references accordingly. [Lines 101, 102, 105, and 108]

L113: Instead of 'conventional' I would rather call this a 'traditionally employed rectangular' grid.

Thank you for your suggestion. We modified it accordingly. [Line 116]

L115: Both the horizontal and the vertical grid spacing of the model are rather coarse. How much may this affect the results? My main concerns would be strat/trop exchange and representation of vertical gradients in the boundary layer. Was this model setup (independently of the present inversion) tested against profile observations?

We thank your comment. In order to answer your concern, we added the following sentences.

"The lowest 12 layers cover the altitude range below about 3 km, with which vertical mixing is reasonably simulated (e.g., see Niwa et al., 2011 for 222Rn). Meanwhile, the vertical grid spacing in the upper troposphere/lower-stratosphere (UT/LS) is relatively coarse (about 1 km), which may cause faster mixing in the UT/LS region. This could affect absolute values of CH4 emission estimates; however, its influence on the results in this study would be limited because temporal variations of $CH_4$ emissions are mainly discussed." [Lines 118–123]

L143: Why not use a monthly factor for the anthropogenic emissions as well? A number of studies have shown strong seasonality in these as well. For example for emissions from natural gas use, which tend to be increased in the cold season when demand is higher.

As pointed out, the anthropogenic emissions have seasonal variations, their magnitudes are, however, one or two orders smaller than those of wetland, fire, and rice emissions. Because we do not perfectly distinguish sectral emissions, we fixed seasonal variabilities of the anthropogenic emissions to the prior estimates.

To clarify this, we added the following sentence.

"This is because, although $f_{anth,i}$ has some seasonal variability, its magnitude is one or two orders smaller than those of fire, wetland or rice emissions." [Line 151–152]

L163: Could the derived prior uncertainties for rice, wetland and soil be given as well. Would be interesting to compare them to the fixed values for anthropogenic sectors.

We added the annualy and globally integrated prior flux errors in Table 1.

Table 1: Last column, last two rows. Instead of N/A consider to repeat the original names (natural, soil).

We modified it accordingly.

L191: Most inversions that use continuous observations from tall towers also apply a temporal filter, assimilating only afternoon observations to avoid known model misrepresentation of boundary layer mixing at other times. Similarly, mountain top observations are often filtered to avoid day-time updrafts. Was such a filter applied here as well?

Yes, we only used data with the large-scale representing flag for the ObsPack datasets. Also, we used daytime data for the JL-STATION. These are added in the manuscript. [Lines 189–190, 195]

Eq. 2, L218f: Please elaborate on this a bit more. I understand that the standard deviations of observations in a certain spatiotemporal area can be used to quantify the model's representative error. But why the expansion with the number of observations? Intuitively, it does not seem to make sense to assign large uncertainties for observations in areas covered by a dense network.  However, the choice of model-data mismatch is very critical for any inversion study. Which is especially true when mixing different kinds of observations as done here.

As pointed out, the choice of model-data mismatch was very critical because we mixed different kinds of observations. To elaborate on this, we added the following sentences.

"The diagonal matrix of $R$ assumes that all observations are independent from each other. However, that is not necessarily the case especially where observations are obtained with high density. Therefore, we inflate the variance for such areas with $N_i$. In fact, Niwa et al. (2022) confirmed that this variance inflation improved inversion results." [Lines 230–233]

L293: Were the continuous in-situ observations not used in the performance analysis or the inversion? It would be useful to see the model performance for these as well. I suppose the performance is much worse than for the flask data, which is expected since the latter will be mostly taken under background conditions and the former are often impacted by recent emission events. Nevertheless, I would urge to show at least prior to posterior improvements to learn if the results are at all comparable to high-resolution regional scale inversions available for Europe and North America. To keep them separate an additional panel for continuous in-situ could be added to Fig 2.

No, the continuous in-situ data were not used in the analysis. As pointed out, it would be better to include continuous in-situ observations to extensively evaluate the inversion performance over North America and Europe. However, we intended to show more global features of the performance in this section. Therefore, we used only flask-air sampling data. Because we have still enough number of flask-air sampling data over Europe and North America as shown in Fig. 1a, this evaluation would not ignore those regions. Nevertheless, we could say almost the same things as noted in the manuscript if Fig. 2 is made including continuous in-situ observations (see the figure below).

[Figure]

Fig. The same as Fig. 2, but all the data including in-situ continuous observations are used for the surface and aircraft evaluations.

In order to clarify that we here evaluate the global features, we modified the text as follows

"…, we evaluated the consistency of posterior atmospheric CH$_4$ mole fractions globally with observations to assess…" [Line 309]

L306ff, Fig. 2: Consider being more specific: Pearson correlation coefficient. How are these stats calculated for the in-situ sites? Pooled for all observations or first by site and then averaged? I would also suggest to include the plot of biases (Fig B1) in Fig 2 as well, since RMSD contains a contribution from the bias and only by showing both one can tell if a large RMSD is due to bias or variability.

In the caption of Fig. 2, we clarified them as Pearson correlation coefficients. [Line 341]

The correlations and RMSDs are calculated from all observations in each latitudinal band. This is clarified in the caption by adding "(directly from all observations in each latitudinal band (with Comilla excluded)" for the surface observation. [Line 344]

We think RMSDs do not contain a contribution from the bias. This is because the averages for 2016–2019 are subtracted in advance (see the last sentence of the caption). To elaborate on this, we added "in each latitudinal band, which would remove the biases shown in Fig. B2 and only show only variations from the reference period of 2016–2019" in the last sentence of the caption. This is already addressed in the main text. [Line 348–349]

In these performance statistics, it is interesting to note, that in the case not discussed in the text (surface observations assimilated) performance against GOSAT observations still largely improves (Fig. 2 c, f), almost similarly well as for GOSAT-based inversions. I think this strongly suggests that GOSAT-based inversions are not fully able to attribute emissions appropriately in the northern extra-tropics, where the surface observations provide the better constraint. With the GOSAT footprint being much wider these emissions are hence allocated elsewhere. Please comment and add to discussion.

As described in Method, the GOSAT-based inversion used the GOSAT observations only. Figure 2 shows that the GOSAT-based inversion could constrain variations (not necessarily absolute values) of $CH_4$ emissions, even though the GOSAT observations may have some biases. Although it is difficult to say that GOSAT could provide further constraints in the northern extra-tropics in addition to those by the surface observations do, GOSAT observations are still useful as independent information from the surface observations. To elaborate on this, we added the sentence below in the main text.

"Furthermore, this result also suggests that GOSAT observations, which may have some biases (Appendix B), could provide constraints to $CH_4$ emission variations as good as those of in-situ and flask observations." [Lines 336–338]

Finally, could these performance statistics be compared with previous global inversions?

As noted earlier, these statistics are specifically made for deviations from the averages for 2016–2019. Therefore, it would be difficult to directly compare them with other inversion studies.

L332: The opposite could be said about the high latitude emissions in Siberia. They are prominent in SURF and SURF+AIR but little changed in GOSAT. For both changes the differences in observational constraint were already mentioned above, but this could be repeated here as well.

As suggested, we added discussion on the differences of the emissions estimated in Siberia. We consider that these latitudinal differences are attributed to the systematic differences between the in-situ or flask observations and the GOSAT observations shown in Appendix B.

Added sentences:

"In the northern high-latitude, the opposite case is true; the SURF and SURF+AIR inversions estimated larger emissions in Siberia than the GOSAT inversion. These latitudinal differences of the estimated emissions are attributable to the systematic differences between the in-situ or flask observations and the GOSAT observations shown in Appendix B." [Lines 358–361]

L371f: Unclear what is meant: the observations are independent in the sense of how they were obtained, but not in the sense of which air masses were sampled. Is the latter, what should be expressed?

We modified as

".., because the in-situ or flask observation and the GOSAT data were independently obtained. The result indicates that those observations consistently captured atmospheric $CH_4$ variations that were likely caused by emission changes." [Lines 407–409]

L390: 'significantly'. In order to judge significance it would be helpful to report posterior uncertainties. See general comment above.

We did not intend to use this word as "satatistically significant", here. Therefore, we changed the word to "clearly". [Line 428]

Section 3.2: The discussion does not cover all regions shown in the Fig 6. I wonder if the last statement about no significant trends should not be extended to include other regions without clear trends (like Oceania, North and Central America).

As suggested, we modified the last sentence as

For other areas such as Europe, the western part of northern Eurasia, Temperate North America, Central America, Southern Africa, Oceania, and Temperate South America, the inversions suggested that $CH_4$ emissions have not clearly contributed to the increase of atmospheric $CH_4$ during 2020–2022. [Lines 427–429]

Fig. 6: Does the gray ribbon actually reflect the prior uncertainty or is it just a thick line? Adding uncertainty ribbons or bars to these time series plots may actually be helpful to judge the significance of the results.

It is just a thick line. As the newly added discussion about the posterior uncertainty values, they are small and cannot be considered as practiaial uncertainties. Therefore, we keep it as is.

L444f: How much could this be a consequence of transport model resolution, prior covariance and/or assigned data-mismatch uncertainty?

We consider that that anti-correlation was caused by atmospheric transport. Prior error covariance has only positive correlations. We think that model errors like those replated to model resolution could not affect correlations among such large regions, especially where observations are densely existing.

To clarify, we added the following sentence.

"Atmospheric transport patterns (such as north to south or south to north winds) might have caused that large anti-correlation." [Lines 485–486]

L459: 'generally consistent'. One very prominent difference between GOSAS and in-situ obs is the shift from wetland to fossil fuel, which for the total was a shift from tropics to high latitudes. Worth mentioning here.

Thank you for pointing out this. We modified the text as

"For the entire period, the GOSAT inversion estimated larger wetland emissions and smaller fossil fuel emissions than the SURF and SURF+AIR inversions, which reflects the larger emissions in the tropics and the smaller emissions in the northern mid-latitudes (Fig. 3). Nevertheless, their temporal changes are…" [Lines 498–501]

Furthermore, we added "and the northern mid-latitude fluxes are smaller" in the description for Fig. 3 to elaborate the shift of emissions from the northern mid-latitudes to the tropics. [Lines 355–356]

L480, Fig. 10: Also the large increase estimated for Southeast Asia (S) seems worth mentioning here, which seems to be compensated by the decrease in biomass burning for the same region and the GOSAT estimates.

In southern Southeast Asia, the large increases of the fossil fuel emissions were not compensated by the decrease of biomass burning emissions. This is because temporal changes of those emissions are not coincident.

The following sentence is added to the text.

"In southern Southeast Asia, the GOSAT inversion estimated larger biomass burning emissions in 2019; however, they diminished in 2020–2022 resulting in the larger decrease of biomass burning $\Delta fCH_4$." [Lines 522–523]

In addition, we modified "Southeast Asia" to "southern Southeast Asia" in the previous sentence. [Line 521]

L489: 'between two'. Actually, correlation pairs for all three sectors are shown. It would be good to show or mention correlation with biomass burning in Souteast Asia as well. See previous comment. There is mentioning of negligible anti-correlations in L506, but is this true everywhere and for correlation with fossil fuel in Southeast Asia (S) in the GOSAT inversion?

We modified as "among the the three sectors". [Line 535]

As mentioned in the above, the fossil fuel emission unlikelly anti-correlates with the biomass burnig emission in Southeast Asia (S). In fact, a neglibile anti-correlation (-0.01) is found for the fossil fuel vs. biomass burning emissions in Southeast Asia (s) in the GOSAT inversion.

Section 4.1: The discussion by region could be more quantitative. How much is the increase in Asian emissions actually reduced if OH is increased? From Fig. C1 is seems clear that the main impact of changed OH is on Tropical African and South American emissions but this is not well

reflected and corroborated by numbers in the text. In addition, the role of continued high emissions in 2021 and 2022 could be mentioned in this context again.

We elaborated Appendix C [Line 727–734] and add the following sentences in the second paragraph of Section 4.1.

"Especially, tropical South America shows the largest emissions reduction of 4 Tg $CH_4$ $yr^{-1}$ with the OH reduction. Meanwhile, the OH reduction induced 2–3 Tg $CH_4$ $yr^{-1}$ emissions reduction in Central Africa, northern and southern Southeast Asia, and South Asia." [Lines 571–573]

Furthermore, we modified the last of the second paragraph as:

"These results indicate that these Asian emissions contributed to the surge of atmospheric $CH_4$ growth from 2019 to 2020. Given the limited OH reduction for 2021–2022 (Liu et al. 2023, Li et al. 2023), those high emissions continued until 2022." [Lines 578–580]

L538: Is there a brief explanation why there are more data in the UoL product? Are there any published results concerning spatial or other biases between the two GOSAT products that would help with the interpretation?

The reason why the NIES product has less data is described in Section 2.3.2 [Lines 244–246] the reason why the UoL product has more data is explained in the second paragraph of Appendix D [Lines 750–752].

Schepers et al. (2012) discussed difference of retrieved $XCH_4$ data between the full-physics method and the proxy method. However, we think that it could not directly help the interpretation at this moment.

To clarify the discussion on this matter, we modified the text as follows:

"the NIES GOSAT product we used" => "the NIES GOSAT product (the full-physics method) we used" [Line 586]

"GOSAT-UoL has" => "the denser data of GOSAT-UoL have" [Line 593]

"In fact, the full physics method and the proxy method could have non-negligible differences in retrieved $XCH_4$ data (Schepers et al., 2012). " is added. [Lines 600–601]

L581: How is the seasonal cycle of posterior high-latitude wetland emissions in the three inversions? We would expect these to peak in summer, when GOSAT observations should be available in the area and pick up a signal of increased column densities. However, we also have large fossil emissions in the same area, which may actually peak in winter (larger demand). GOSAT would have trouble to notice these emissions (no sunlight). Fig. 11 suggests that fossil and wetland emissions are not well separated by the inversion for Northern Eurasia (W). How much could seasonal misattribution and seasonal lack of observations contribute to the discrepancies between SURF and GOSAT inversions?

Definitely, the seasonal sampling difference of GOSAT is an interesting issue to investigate. However, we think that that problem is out of the scope of this study, which focus on year-to-year variations especially for the rapid growth for 2020-2022. We would like to investigate this issue in a future study.

Nevertheless, in Nortthern Eurasia (W), we confirmed that the GOSAT inversion produced seasonal variations more similar to those of the SURF inversion than to the prior ones even for winter, suggesting some constraints of GOSAT there.

L594f: How would one explain a sharp increase in emissions from the agricultural & waste sector from one year to the next? Usually changes in these sectors are slow. It seems more likely that wetlands were the sole driver and the inverse method is not capable of separating them fully, as indicated by the negative posterior correlations.

We agree that wetland emissions are more likely to be the main driver of the emission growth. However, a growth of rice cultivation emissions, one of anthropogenic emissions, cannot be denied as well, because they are remarkably large in those regions.

For either the prior or posterior data, rice cultivation emissions are the largest in northern Southeast Asia (approximately twice as large as wetland emissions) and comparable to wetland emissions in South Asia.

Therefore, we modified as

"Given that changes of anthropogenic emissions are slow, it seems likely that wetlands were the main driver of the emission growth. However, for either the prior or posterior data, rice cultivation emissions are the largest in northern Southeast Asia (approximately twice as large as wetland emissions) and comparable to wetland emissions in South Asia, a growth of such anthropogenic emissions cannot be denied and it would suggest a potential impact of direct emissions reduction measures on this sector for these two Asian regions." [Lines 650–654]

L601: 'agree with each other'. To me this statement is too abbreviated as they do not fully agree in the spatial and sectorial attribution.

We modified as

"the inversions with independent observations (SURF(or SURF+AIR) and GOSAT) agree in finding that biogenic emissions in the tropics and northern low-latitudes are the main contributors to the emissions increases. " [Lines 656–658]

L604: 'newly introduced'. Were these data not used in any other global inverse modelling study before? Maybe reword to emphasize this fact.

To our knowledge, other modeling studies did not use such a dense observation network in Asia. Especially, the use of the mobile observations, such as ship and aircraft observations, is uniqueness of our study. To emphasize this, we modified the text as

"Furthermore, the dense observation network including not only surface observations but also ship and aircraft observations, which was newly introduced in this study, provided strong constraints and increased the confidence in the Asian flux estimates." [Lines 660–662]

L606ff: Another conclusion from the study is the need for unbiased satellite products and inversions combining surface in-situ, aircraft and satellite observations.

Because we could not conclude whether the spatial coverage difference or biases in the satellite products induced those different flux estimates, we tink the need for unbiased satellite products cannot be mentioned here. The need of in-situ and flask observations we mentioned here is to evaluate those different satellite-based flux estimates. To clarify this, we modified the text as follows.

"To evaluate those different satellite-based estimates, we need more elaborate networks of high-precision in-situ and flask observations not only at the surface but also in the upper-air (by aircraft); these observations are specially needed in the tropical and low-latitude areas of Africa, South America, and Asia." [Lines 665–668]

Appendix B: I suggest to integrate Fig. B1 and the discussion of bias in the main text. See comment above.

As replied above, Figure 2 excluded contributions from the biases shown in Fig. B2, showing different information from Fig. B1. Therefore, we keep Appendix B as it is.

Technical comments

L82f: Consider 'spatial coverage is limited' instead.

Thank you for your suggestions. We modified it accordingly. [Lines 84–84]

L83: No 'the' in front of 'low latitudes'. No hyphen in the latter either. 'remain poorly covered by' instead of 'remained poor in'.

Thank you for your suggestions. We modified them accordingly. [Lines 85–86]

L85: Consider 'conditions' instead of 'areas'.

Thank you for your suggestions. We modified it accordingly. [Line 87]

L110: Repeated use of 'transport model'. Would '… adjoint tracer transport model of NICAM-TM (Niwa et al., 2011, 2017b).' work as well?

Because NICAM-TM first appears here, we cannot abbreviate it. Instead, we changed "adjoint tracer transport models of" to "adjoint modes of".  [Line 113]

Citation: https://doi.org/10.5194/egusphere-2024-2457-RC2